# CAN TRANSFORMERS DO ENUMERATIVE GEOMETRY?

**Baran Hashemi**
ORIGINS Data Science Lab
Technical University Munich
`baran.hashemi@tum.de`

**Roderic G. Corominas**
Department of Mathematics
Harvard University
`rguigo@math.harvard.edu`

**Alessandor Giacchetto**
Departement Mathematik
ETH Zürich
`alessandro.giacchetto@math.ethz.ch`

## ABSTRACT

We introduce a Transformer-based approach to computational enumerative geometry, specifically targeting the computation of $\psi$-class intersection numbers on the moduli space of curves. Traditional methods for calculating these numbers suffer from factorial computational complexity, making them impractical to use. By reformulating the problem as a continuous optimization task, we compute intersection numbers across a wide value range from $10^{-45}$ to $10^{45}$. To capture the recursive nature inherent in these intersection numbers, we propose the Dynamic Range Activator (DRA) [1], a new activation function that enhances the Transformer's ability to model recursive patterns and handle severe heteroscedasticity. Given the precision required to compute these invariants, we quantify the uncertainty in the predictions using Conformal Prediction with a dynamic sliding window, adaptive to partitions of equivalent numbers of marked points. To the best of our knowledge, there has been no prior work on modeling recursive functions with such a high-variance and factorial growth. Beyond simply computing intersection numbers, we explore the enumerative "world-model" of Transformers. Our interpretability analysis reveals that the network is implicitly modeling the Virasoro constraints in a purely data-driven manner. Moreover, through abductive hypothesis testing, probing, and causal inference, we uncover evidence of an emergent internal representation of the the large-genus asymptotic of $\psi$-class intersection numbers. These findings suggest that the network internalizes the parameters of the asymptotic closed-form and the polynomiality phenomenon of $\psi$-class intersection numbers in a non-linear manner.

## 1 INTRODUCTION

Enumerative geometry is a branch of mathematics concerned with counting geometric objects or finding invariants that satisfy certain geometric conditions (Mumford, 1983; Katz, 2006). A classical question in enumerative geometry is: *How many conics pass through five given points in the plane?* A more contemporary one is: *How many rational curves of degree one are there on a quintic threefold?* Addressing such problems often boils down to computing intersection numbers, which are fundamental in computing Weil–Petersson volumes, Hurwitz numbers, Gromov–Witten invariants, and quantum gravity amplitudes. An example of such objects is $\psi$-class intersection numbers. Traditional recursive methods like the KdV hierarchy (Witten, 1991; Kontsevich, 1992) have com-

---

[1]GitHub Code: `https://github.com/Baran-phys/DynamicFormer`

putational complexities that grow like $O(g! \cdot 2^n)$, where $g$ is the genus and $n$ is the number of marked points, making calculations infeasible even for low values of $g$ and $n$. To overcome this computational bottleneck, we propose employing Machine Learning (ML), particularly Transformers (Vaswani et al., 2017), to approximate $\psi$-class intersection numbers.

It is well-known that Transformer-based models have difficulty with even periodic patterns, let alone recursive problems (Ziyin et al., 2020). Therefore, we first examine the architectural design choices and training pipeline variants necessary for learning the recursive reasoning processes involved in quantum Airy structures (Kontsevich & Soibelman, 2018; Andersen et al., 2024). Hence, we introduce a non-linear activation function, the Dynamic Range Activator (DRA), which is particularly suited for learning recursive functions. Consequently, we develop DynamicFormer, a modified Transformer-based model designed to predict $\psi$-class intersection numbers given the quantum Airy structure initial data. Additionally, we believe that a predictive model without uncertainty estimation is inherently unreliable. Therefore, we incorporate uncertainty quantification into our model's predictions using Conformal Prediction (Shafer & Vovk, 2008). Going further, we investigate whether the network is capable of abductive reasoning for knowledge discovery. Specifically, we ask whether we can conduct parametric inference and extract mathematical insights from its internal understanding beyond the raw data provided to offer intuition and potentially generate new conjectures. We provide causal and correlational evidence that the network is actually understanding the underlying mathematics. As a result, we aim to introduce a new paradigm for using Transformers in algebraic geometry as both a powerful and explainable amortization method and a source of intuition for the mathematician.

While the studies discussed in Appendix B primarily focus on symbolic logic or compact numerical sequences, they fall short in handling the highly recursive and complex structures found in enumerative geometry. Existing models are limited to in-distribution settings and struggle with the high-variance and recursive nature of problems in computational algebraic geometry. For example, Belcak et al. (2022) discuss and benchmark various ML models only up to unique-type functions and do not even enter the realm of recursive functions.

Our work bridges machine-human collaborative guidance and predictive problems while introducing new methods. Most studies in prediction tasks remain within in-distribution and compact settings, whereas we extend our approach to out-of-distribution recursive predictions, incorporating Conformal Prediction uncertainty estimation—an often overlooked aspect in the literature. Precision is essential in mathematics, especially when a mathematician needs to assess whether an informed conjecture is correct based on available samples. In such decision-making processes, having access to the uncertainty of AI-predicted samples/values is crucial, as it provides insight into the reliability of the predictions and guides the validation of mathematical hypotheses. We also enhance the predictive task by using multi-modal data, involving both discrete sets and continuous graph (sequence) representations. This approach moves beyond the focus on mere compact symbolic/numerical sequence modalities (Meidani et al., 2024). Furthermore, in the field of machine-human collaborative guidance, we conduct causal inference and correlational analyses to understand the mathematical "world-model" of Transformers (Li et al., 2023; Micheli et al., 2023), offering deep insights into the underlying algebraic geometry problem at hand.

To the best of our knowledge, this is the first time Transformers and explainable ML have been applied to enumerative algebraic geometry to extract knowledge and illuminate the model's internal understanding of deep research-level mathematical concepts. In the end, we aim to target both machine learning scientists and mathematicians, as our work is relevant to both communities.

## 2 QUANTUM AIRY STRUCTURES

The conceptually simplest approach to computing $\psi$-class intersection numbers involves using Virasoro constraints recast in the language of quantum Airy structures (Kontsevich & Soibelman, 2018; Andersen et al., 2024). The latter is an algebraic reformulation of topological recursion (Eynard & Orantin, 2007). A quantum Airy structure on a complex vector space $V$, dually generated by $(x^i)_{i \in I}$, is a family of differential operators $(L_i)_{i \in I}$ in the Weyl algebra over $V$ of the form

$$L_i = \hbar \partial_i - \hbar^2 \sum_{a,b \in I} \left( \frac{1}{2} A_{i,a,b}\, x^a x^b + B^b_{i,a}\, x^a \partial_b + \frac{1}{2} C^{a,b}_i\, \partial_a \partial_b \right) - \hbar^2 D_i \,, \tag{2.1}$$

and forming a Lie sub-algebra of the Weyl algebra. Here $\partial_i = \partial_{x^i}$ and $A_{i,a,b} = A_{i,b,a}$, $B_{i,a}^b$, $C_i^{a,b} = C_i^{b,a}$, and $D_i$, are scalars indexed by elements in $I$, and $\hbar$ is a formal parameter that keeps track of the genus. Given a quantum Airy structure, one can uniquely find a formal function $Z$ on $V$, called *partition function*, which is annihilated by the differential operators $(L_i)_{i \in I}$. More precisely, the following theorem holds.

**Theorem 2.1.** *If $(L_i)_{i \in I}$ is a quantum Airy structure on $V$, there exists a unique formal function $Z$ of the form*

$$Z(\boldsymbol{x}; \hbar) = \exp\left( \sum_{\substack{g \geq 0 \\ n > 0}} \frac{\hbar^{2g-2+n}}{n!} \sum_{d_1,\dots,d_n \in I} F_{g;d_1,\dots,d_n} \, x^{d_1} \cdots x^{d_n} \right) \tag{2.2}$$

*with $F_{0;d_1} = F_{0;d_1,d_2} = 0$, $F_{g;d_1,\dots,d_n}$ symmetric in $d_1, \dots, d_n \in I$, and such that*

$$L_i Z = 0 \qquad \forall i \in I. \tag{2.3}$$

The elements $F_{g,n} = \sum_{d_1,\dots,d_n \in I} F_{g;d_1,\dots,d_n} \, x^{d_1} \cdots x^{d_n}$ are symmetric tensors of rank $n$ over $V^*$. The collection of coefficients $(A, B, C, D)$ can also be regarded as tensors, and we will often refer to them as the quantum Airy structure initial data. For a given quantum Airy structure, the coefficients $F_{g;d_1,\dots,d_n}$, called *quantum Airy structure amplitudes*, are determined recursively on (minus) the Euler characteristic $2g - 2 + n$. For the lowest values of $g$ and $n$, the base cases are given by $F_{0;i,j,k} := A_{i,j,k}$ and $F_{1;i} := D_i$, where $A_{i,j,k}$ and $D_i$ are part of the data of the quantum Airy structure. For higher values of $g$ and $n$ satisfying $2g - 2 + n > 0$, the recursion formula, known as *topological recursion*, is given by

$$F_{g;d_1,d_2,\dots,d_n} = \sum_{m=2}^n \sum_{a \in I} B_{d_1,d_m}^a F_{g;a,d_2,\dots,\widehat{d_m},\dots,d_n}$$

$$+ \frac{1}{2} \sum_{a,b \in I} C_{d_1}^{a,b} \left( F_{g-1;a,b,d_2,\dots,d_n} + \sum_{\substack{g_1+g_2=g \\ I_1 \sqcup I_2 = \{d_2,\dots,d_n\}}} F_{g_1;a,I_1} \, F_{g_2;b,I_2} \right). \tag{2.4}$$

In general, the computation of the amplitudes has a computational complexity of $O(g! \cdot 2^n)$, which makes the calculation of the amplitudes at high genera problematic. For high-dimensional vector spaces $V$, finding a closed-form solution becomes increasingly impractical, if not impossible. The central example is the one associated to the $\psi$-class intersection numbers. In this case, the underlying vector space is generated by vectors indexed by $I = \mathbb{N}$, the non-negative integers. The differential operators $(L_i)_{i \geq 0}$ are determined by the tensors

$$A_{i,j,k} := \delta_{i,j,k}, \qquad\qquad B_{i,j}^k := \delta_{i+j,k+1} \frac{(2k+1)!!}{(2i+1)!!(2j-1)!!},$$

$$C_i^{j,k} := \delta_{i,j+k+2} \frac{(2j+1)!!(2k+1)!!}{(2i+1)!!}, \qquad\qquad D_i := \frac{\delta_{i,1}}{24}. \tag{2.5}$$

The resulting operators, after shifting of the indices and rescaling as $\mathsf{L}_i = -\frac{(2i-1)!!}{2} L_{i-1}$, form a representation of the Virasoro algebra: $[\mathsf{L}_i, \mathsf{L}_j] = \hbar^2 (i-j) \mathsf{L}_{i+j}$ for all $i \geq -1$. Remarkably, the associated amplitudes coincide with $\psi$-class intersection numbers (see Appendix C):

$$F_{g;d_1,\dots,d_n} = \begin{cases} \langle \tau_{d_1} \cdots \tau_{d_n} \rangle_{g,n} & \text{if } d_1 + \cdots + d_n = d_{g,n} := 3g - 3 + n, \\ 0 & \text{otherwise.} \end{cases} \tag{2.6}$$

In this paper, we focus on the specific choice of quantum Airy structure data given by Equation (2.5), thereby computing the $\psi$-class intersection numbers. For ease of notation, we will denote a generic partition of $d_{g,n}$ of length $n$ as $\boldsymbol{d} = (d_1, \dots, d_n)$. The associated $\psi$-class intersection number will be denoted by $\langle \boldsymbol{d} \rangle_{g,n} := \langle \tau_{d_1} \cdots \tau_{d_n} \rangle_{g,n}$.

The goal of the next sections is to provide the possibility of regressing these numbers via a Transformer-based model, given the quantum Airy structure initial data. Our model is trained on

known data computed by a brute force algorithm up to genus 13 and is tested up to genus 17. During experimentation, we observed that when comparing the contributions of $B$ and $C$, the $B$ tensor had a greater impact on $\psi$-class intersection numbers. Consequently, we incorporated only this initial datum (excluding $C$), a decision further supported by the inherent properties of the intersection numbers (see Appendix D). This initial observation motivated the subsequent series of explainability analysis experiments and findings.

## 3    METHODS

The recursive nature of functions in enumerative geometry, combined with the sparse and high-variance target distributions of $\psi$-class intersection numbers, introduces a high complexity in modeling and accurately approximating these functions. A main property of recursive maps (e.g factorial function), is their dramatic growth and drop. Learning this recursive behavior requires not only fitting high-frequency patterns within a bounded region but also successfully extrapolating those patterns beyond that region. In time series prediction tasks, capturing periodic even behavior is a challenge. Various methods (Miller et al., 2024) have been employed to model periodic patterns effectively. However, these approaches typically deal with uni-modal data that also exhibit relatively low variance in both In-Distribution (ID) and Out-Of-Distribution (OOD) regions and do not generalize well to recursive problems with the high-variance observed in our context. Therefore, traditional methods for modeling high-variance recursive data, including those used in time series analysis and standard Transformer architectures, are insufficient for our needs (Lakretz et al., 2021; Belcak et al., 2022; Zhang et al., 2024). Thus, to capture such behavior and perform proper inference for multi-modal recursive problems, we enhance Transformers by introducing the Dynamic Range Activator (DRA), and introduce DynamicFormer, depicted in Figure 5. The DRA is designed to handle the recursive and factorial growth properties inherent in enumerative problems with minimal computational overhead and can be integrated into existing neural networks without requiring significant architectural changes.

**Dynamic Range Activator.** It has been shown (Parascandolo et al., 2017; Dauphin et al., 2017; Ziyin et al., 2020; So et al., 2021) that the choice of activation functions plays an important role in the interpolation and extrapolation properties of Transformers. To enable Transformer models to precisely capture the recursive behavior of the data, we introduce a simple activation function that we call the Dynamic Range Activator (DRA). DRA integrates both harmonic and hyperbolic components as follows,

$$\mathrm{DRA}(x) \coloneqq x + a\sin^2\left(\frac{x}{b}\right) + c\cos(bx) + d\tanh(bx)\,, \tag{3.1}$$

where $a, b, c, d$ are learnable parameters. It allows the function to simultaneously model periodic data (through sine and cosine) and rapid growth or attenuation (through the hyperbolic tangent) response. DRA is inspired by the Snake non-linear function (Ziyin et al., 2020). However, Snake only offers a sinusoidal modulation added to a linear term, which, while providing a basic non-linear transformation, lacks the additional flexibility for rapid damping or amplification effects that hyperbolic tangents can provide.

To demonstrate the advantage of DRA in capturing recursive behavior, we set up a small experiment. We generate a small dataset based on the recursive function $r(n) = n + (n, \mathrm{AND}, r(n-1))$, where AND is the bitwise logical AND operator (Sloane, 2007). We train a fully connected neural network with two hidden layers consisting of 64 and 32 neurons over the interval $n \in [0, 120]$, then test the model over $n \in [121, 200]$. we also provide a comparison between Multi-Layer Perceptron (MLP) and the vanilla Kolmogorov–Arnold Networks (KAN) (Liu et al., 2024). The results are shown in Figure 1. This experiment demonstrates that ReLU, Tanh, Gated Linear Unit (GLU) (So et al., 2021), and KAN are unable to capture the recursive nature of the underlying function within a finite training time. Snake appears to learn some periodicity in both the training and testing regions. In contrast, DRA shows a better ability to capture the correct fluctuations, as well as the rapid growth and drops of the underlying recursive function, in both the interpolation and extrapolation regimes. This provides an evidence that DRA has the desired flexibility for modeling recursive behavior and has the potential to effectively model such problems. We further conduct experiments, in Section 4, to demonstrate the advantages of DRA over typical non-linear activation functions in predicting $\psi$-class intersection numbers.

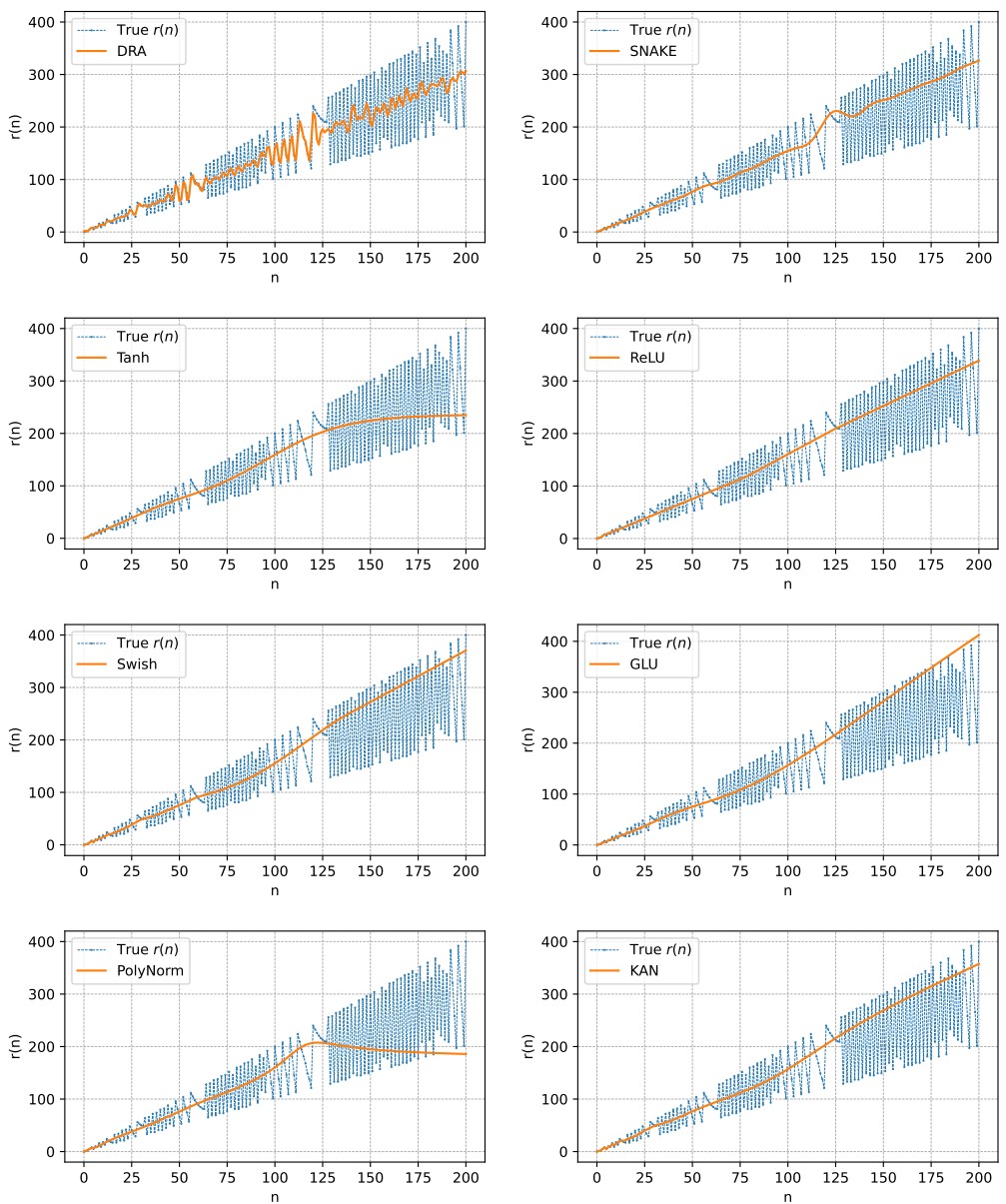

Figure 1: Comparison of DRA MLP with Snake, ReLU, Tanh, GLU, Swish, and PolyNorm activation functions, and KAN in training region $n \in [0, 120]$ and test (extrapolative) region $n \in [121, 200]$. We ensured that all models have a comparable number of parameters. The MLP with DRA non-linearity demonstrates superior performance in capturing the function's behavior.

# 4  RESULTS

To evaluate the performance, we consider two setups: In-Distribution (ID) and Out-Of-Distribution (OOD) settings. We use $R^2$ to compare the results and assess the goodness-of-fit in the intersection number regression task. We then compare DRA with ReLU, Snake, and GLU activation functions as baselines.

### 4.1 IN-DISTRIBUTION RESULTS

In the ID setting, we examine data with the same genera as the training data, that is $g_{\text{ID}} = [1, 13]$, but with different, unseen numbers of marked points $n_{\text{ID}} \in [35, 11]$. In this setting, the $R^2$, the empirical coverage, and the conformal width (CW) (see Appendix G) are shown in Table 1a.

| $(g, n)$ | $R^2 \uparrow$ | Coverage | CW |
|---|---|---|---|
| $(1, 35)$ | 99.8 | 90.35 | 1.03 |
| $(2, 33)$ | 99.6 | 83.60 | 0.84 |
| $(3, 31)$ | 99.9 | 74.79 | 0.76 |
| $(4, 29)$ | 98.7 | 95.66 | 1.11 |
| $(5, 27)$ | 99.1 | 92.66 | 1.03 |
| $(6, 25)$ | 99.3 | 91.18 | 0.80 |
| $(7, 23)$ | 99.1 | 93.05 | 0.68 |
| $(8, 21)$ | 99.8 | 90.88 | 0.76 |
| $(9, 19)$ | 99.9 | 96.71 | 0.91 |
| $(10, 17)$ | 99.9 | 90.01 | 1.04 |
| $(11, 15)$ | 99.8 | 91.97 | 0.87 |
| $(12, 13)$ | 99.6 | 89.08 | 1.30 |
| $(13, 11)$ | 99.9 | 95.90 | 0.97 |

(a) $R^2$ and conformal uncertainty estimation results with $\alpha = 0.1$ (90% target coverage) in the ID setting.

| $(g, n)$ | $R^2 \uparrow$ | Coverage | CW |
|---|---|---|---|
| $(14, [1, 9])$ | 99.6 | 93.82 | 0.93 |
| $(15, [1, 7])$ | 95.9 | 84.27 | 0.91 |
| $(16, [1, 5])$ | 94.1 | 89.60 | 3.55 |
| $(17, [1, 3])$ | 93.8 | 95.27 | 8.30 |

(b) $R^2$ and conformal uncertainty estimation results with $\alpha = 0.1$ (90% target coverage) in the OOD setting.

Table 1: $R^2$ and conformal uncertainty estimation results.

### 4.2 OUT-OF-DISTRIBUTION RESULTS

Now, we study the more challenging task of OOD prediction. In the OOD setting, we examine data with a higher genera than the training data, specifically $g_{\text{OOD}} = [14, 15, 16, 17]$, and a number of marked points $n_{\text{OOD}} \in [1, 9]$. In this setting, the $R^2$, the empirical coverage, and the conformal width are shown in Table 1b. As evidenced by Table 2, the DRA outperforms the ReLU, GLU, and Snake activation functions in predicting the recursive intersection numbers in the OOD setting.

| ReLU | | GLU | | Snake | | DRA | |
|---|---|---|---|---|---|---|---|
| $R^2 \uparrow$ | $CW \downarrow$ | $R^2 \uparrow$ | $CW \downarrow$ | $R^2 \uparrow$ | $CW \downarrow$ | $R^2 \uparrow$ | $CW \downarrow$ |
| 71.5 | 9.73 | 74.7 | 8.34 | 82.9 | 6.55 | **95.8** | **3.42** |

Table 2: Comparison of $R^2$ and Conformal Width between models with ReLU, GLU, Snake, and DRA as their activation functions in the OOD regime.

## 5 HOW DOES THE NETWORK DO ENUMERATIVE GEOMETRY?

Up until now, we have tried to showcase that DynamicFormer is capable of predicting $\psi$-class intersection numbers. However, several key questions remain. *Do Transformers actually understand the underlying enumerative geometry? Is it possible to extract useful knowledge or hints toward a possible (re)discovery of a theorem?* To address these questions, we have made several interesting observations throughout our work that offer deeper insights into the network's "world model" and learning process, hinting at potential mathematical knowledge discovery.

### 5.1 THE DILATON EQUATION

The topological formula equation 2.4 for the specific case of $\psi$-class intersection numbers and the choice of $d_1 = 1$ takes a particularly simple form, known as the Dilaton equation,

$$\langle 1, d_1, \ldots, d_n \rangle_{g,n+1} = (2g - 2 + n) \langle d_1, \ldots, d_n \rangle_{g,n}. \tag{5.1}$$

The equation states that the intersection number involving a $\tau_1$-term can be reduced to a simpler intersection number with one fewer marked point.

Let $\langle d \rangle_{g,n}$ be the intersection numbers indexed by genus $g$ and number of marked points $n$. Given the the set of values of the tensor $B$ and the set of partitions $d \in \mathbb{N}^n$, the neural network embedding $p_{g,n} \colon \{ \mathbb{R}^{d_{g,n} \times d_{g,n} \times d_{g,n}}, \mathbb{N}^n \} \to \mathbb{R}^k$ maps the input data to a $k$ dimensional vector space $\mathbb{R}^k$ denoted as $\boldsymbol{x}_{g,n}(\langle d \rangle_{g,n} \mid B, d)$. This embedding describes the hidden understanding of the Transformer just before the prediction head, capturing the learned representation of the intersection numbers. By equipping $\mathbb{R}^k$ with the standard inner product, $S_{i,j} : \mathbb{R}^k \times \mathbb{R}^k \to \mathbb{R}$, we obtain the cosine of the angle between the normalized embeddings of $\psi$-class intersection numbers as a measure of geometric similarity:

$$S_{i,j} = \frac{\langle x_{g,n}^i, x_{g,n}^j \rangle}{\|x_{g,n}^i\| \, \|x_{g,n}^j\|} \,. \tag{5.2}$$

Observing an interesting recursive pattern in Figure 2, we hypothesize that they stem from the Virasoro constraints between intersection numbers. To verify this, we visualized the Dilaton equations and confirmed that the model has rediscovered it as relationships between intersection numbers. We translated the Dilaton equation into a Dilaton relation matrix by flagging any two intersection numbers that satisfy this equation. We then observed that the intersection numbers with high cosine similarity follow at least the Dilaton equation. We anticipate that other observed patterns may also be manifestations of the remaining Virasoro constraints. This phenomenon persists even in the OOD setting. This explains how the model has succeeded in generalizing well by learning the underlying symmetries and relations. It provides an evidence that the network has learned the Virasoro constraints in a data-driven way without prior exposure to these governing rules.

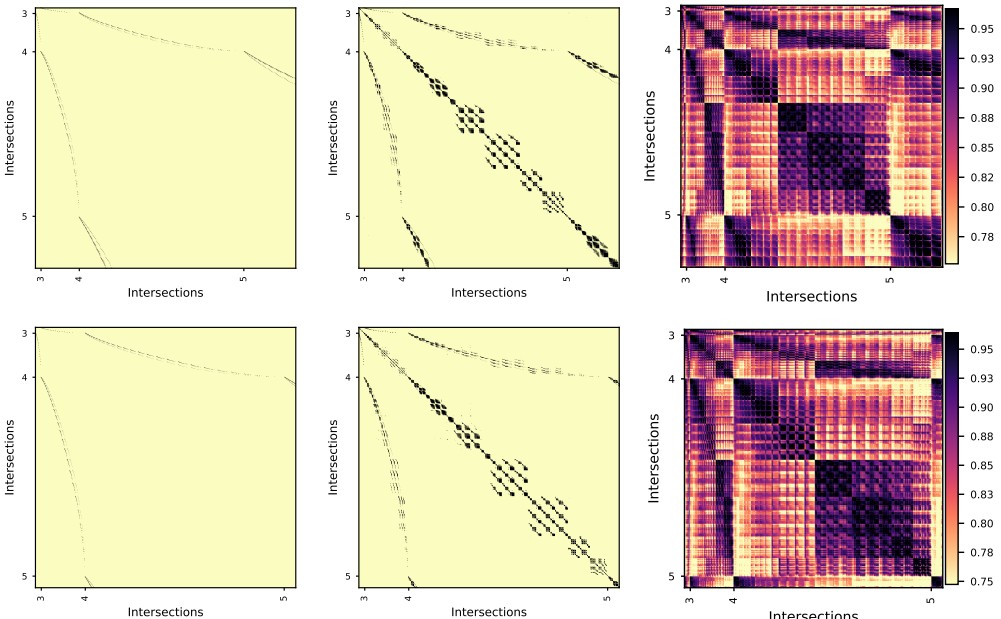

Figure 2: The Dilaton relations matrix (left) for $\psi$-class intersection numbers at $g = 13$(top) and $g = 14$(bottom). Cosine similarity between embeddings of predicted intersection numbers (right) and with a cut $S_{i,j} \geq 0.98$ (middle).

### 5.1.1 LARGE GENUS ASYMPTOTIC AND ABDUCTIVE REASONING

Delecroix et al. (2021) conjectured that, in the large genus limit, $\psi$-class intersection numbers simplify to a specific asymptotic form.

**Theorem 5.1.** *For $n = o(\sqrt{g})$, uniformly in $d_1, \ldots, d_n$ as $g \to +\infty$:*

$$\langle \boldsymbol{d} \rangle_{g,n} \prod_{i=1}^n (2d_i + 1)!! = \frac{2^n}{4\pi} \frac{\Gamma(2g - 2 + n)}{(\frac{2}{3})^{2g-2+n}} \left( 1 + o(1) \right). \tag{5.3}$$

*Here $\Gamma(x)$ is the Euler Gamma function.*

This theorem, proven by Aggarwal (2021), illustrates that, as the genus $g$ grows large, the intersection numbers grow factorially as $(2g)!$ with a specific exponential behavior modulated by the constant $A = 2/3$. Recently, another approach for computing the large genus asymptotics of intersection numbers was proposed by Eynard et al. (2023). The strategy of this proof is based on a resurgent analysis of the $n$-point functions of $\psi$-class intersection numbers, which are computed via determinantal formulae. The determinantal formula is a consequence of the integrability property of the intersection numbers, namely KdV. With this approach, the authors extended Aggarwal's results by computing all subleading corrections:

$$\langle \boldsymbol{d} \rangle_{g,n} \prod_{i=1}^{n} (2d_i + 1)!! = \frac{2^n}{4\pi} \frac{\Gamma(2g-2+n)}{(\frac{2}{3})^{2g-2+n}} \left( 1 + \frac{\frac{2}{3}\alpha_1}{2g-3+n} + \cdots \right). \qquad (5.4)$$

**Abductive Reasoning:** Identifying components and parameters such as $A = 2/3$ or the subleading corrections in Equation (5.4) can also be approached through abductive reasoning, framing the task as hypothesis testing. Abduction involves proposing plausible explanations for observed data, to determine which best aligns with the network's understanding of the data. In this context, we investigate how the network can infer the parameters like the constant $A = 2/3$ in the asymptotic formula, treating it as an inverse problem. Thus, an important question we want to answer is: *In scenarios where parameters in asymptotic formulas are unknown, can we use abduction to infer and provide evidence for their potential values?* To address this, we first examine the reasoning of DynamicFormer by deciphering its understanding of the asymptotic parameters of the intersection numbers, which are not explicitly provided to the model. The parameters we aim to infer and rediscover are the exponential growth $A$ and the first few subleading asymptotic polynomials $\alpha_1, \alpha_2, \ldots$ (Guo & Yang, 2022; Eynard et al., 2023).

One can claim that there is an evidence that the model's embedding vector space $\boldsymbol{x}_{g,n}(\langle \boldsymbol{d} \rangle_{g,n} | \boldsymbol{B}, \boldsymbol{d})$ encodes the parameter $A = 2/3$ and the subleading polynomials if there is a correspondence between the embeddings and these parameters. A prominent method that attempts to shed light on this correspondence is *probing* (Conneau et al., 2018), also known as diagnostic prediction (Hupkes et al., 2020). Under this methodology, one trains a linear or non-linear model as a probe to predict any desired information from the latent representations of the network. Formally, we aim to evaluate how well the network's hidden representations encode the fundamental parameter $A$ by predicting the target function $I_j$ associated with each input $j$. Let $\boldsymbol{x}_{g,n}^j \in \mathbb{R}^k$ be the hidden state for input $j$, and $I_j$ be defined based on approximate conjectural estimation of Equation (5.4). To map the hidden representations $\boldsymbol{x}_{g,n}^j$ to the target values $I_j$, we employ both linear and non-linear (MLP) probes $f$. The probe $f$ is trained, using a conformal prediction procedure (Shafer & Vovk, 2008) to quantify the statistical uncertainty, by minimizing the squared error between the predicted and actual values of $I_j$, $f = \arg\min_\theta \sum_j (f_\theta(\boldsymbol{x}_{g,n}^j) - I_j)^2$. High prediction performance is interpreted as an evidence that the information is encoded in the Transformer's enumerative world model. The efficacy of the encoding is gauged by the coefficient of determination ($R^2$) of the probe.

For $A = 2/3$, we set up the experiment with a grid of alternative hypotheses. Specifically, we probe a discrete value space for the numerator of $A$, $\mathrm{num} = 1, \ldots, 10$, and the denominator, $\mathrm{denom} = 1, \ldots, 10$, which covers 90 candidates (with $A = 1$ excluded as a trivial hypothesis). It is expected from the Wentzel–Kramers–Brillouin (WKB) method applied to the underlying Airy quantum curve that $A$ would be a period on the associated Riemann surface. For this specific enumerative problem, periods are in $\mathbb{Q}$. We use both linear and non-linear probes to see if the network's internal representation for $A = 2/3$ has a linear form or not. The higher and more precise performance of the non-linear probe to the linear one suggests that $A$ could be encoded non-linearly, as shown in Figure 3. We perform the same analysis for the first few terms of the subleading series $\alpha_k$ with $k = 0, 1, 2, 3$. We found that with $90\%$ coverage, the Transformer's vector space representation can predict the coefficients with an $R^2 = 0.98$. Notably, the linear probe yields a lower value $R^2 = 0.63$. **This suggests that the network's internal representation of the polynomiality phenomenon (Guo & Yang, 2022; Eynard et al., 2023) does not have a simple linear form, but a non-linear representation.**

**Causal Tracing:** Originally, causal tracing (Vig et al., 2020) was introduced to quantify information storage and transfer within Transformer components (Meng et al., 2022; Feng & Steinhardt, 2024;

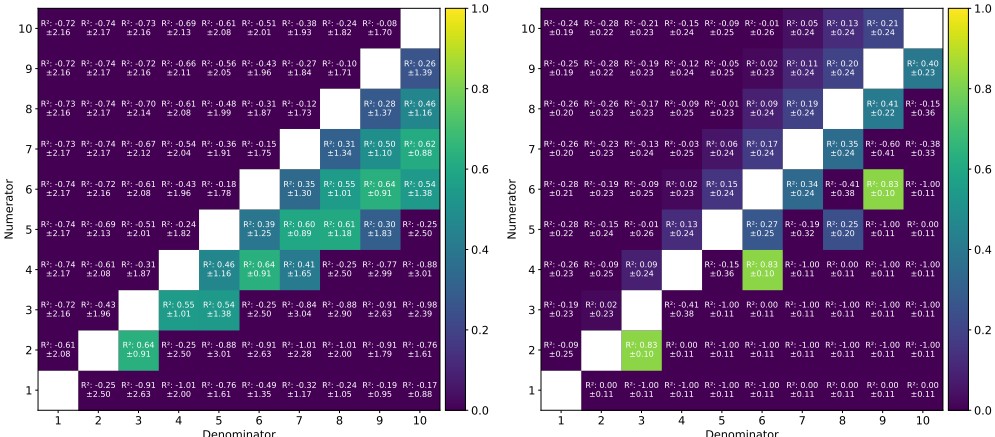

Figure 3: The network's internal representation's linear (left) and non-linear (right) predictive power of the exponential growth constant of $\psi$-class intersection numbers.

Wang et al., 2024), which is not our focus here. Instead, we use causal tracing to analyze the model's decision-making, specifically identifying which input modalities influence the prediction of $\psi$-class intersections. In other words, we aim to uncover how the network causally comprehends the underlying mathematics. We define an input modality as $m[i] = \{ n[i], \boldsymbol{B}[i], \boldsymbol{d}[i] \}$ for each instance $i$. There are then two steps:

1. **Clean run.** This run records the model's prediction on a regular input. With this run, we record auto-correlation of the predicted intersection numbers across different number of marked points.

2. **Counter-factual intervention.** We perform counterfactual interventions (Baron, 2023) by modifying instances in one modality while keeping others unchanged to observe how these changes affect the model's predictions. This involves incorrectly assigning features to instances—for example, associating the partition $\boldsymbol{d} = (29, 10, 5, 3, 1, 0)$ with $n = 3$ instead of $n = 6$. Previous research on factual associations in Transformers has explored methods such as adding noise to inputs (Meng et al., 2022), which creates redundant distribution shifts (Zhang & Nanda, 2024), and exchanging tokens (Feng & Steinhardt, 2024). Our approach is similar to the latter method, focusing on how changes in input tokens affect model predictions. In our case, for the modality of interest (while keeping the other modalities intact), we replace the input instances with random alternatives of the same genus. This replacement results in a miss-assignment of features to a sample, which alters the target prediction, i.e., $m[i] \rightarrow m[j]$ where $i \neq j$, to obtain $\boldsymbol{x}_{g,n}(\langle \boldsymbol{d} \rangle_{g,n} | \mathrm{do}(m[i]))$. This allows us to study how the model relates features to instances and to attribute cause and effect between interventions and the intersections. The difference observed between the clean and intervened runs is measured by $\mathrm{R}^2$, the autocorrelation of the intersection numbers, and the $\mathrm{R}^2_{\mathrm{probe}}$ of the linear probe for the exponential growth constant $A = 2/3$.

|  | No Intervention | $n$ | $\boldsymbol{B}$ | $\boldsymbol{d}$ |
|---|---|---|---|---|
| $\mathrm{R}^2$ | 0.96 | $-12.6$ | 0.54 | $-52.2$ |
| $\mathrm{R}^2_{\mathrm{probe}}$ | 0.83 | 0.52 | $-2.77$ | 0.43 |

Table 3: Causal Strength for Different Modalities

**What causal reasoning is the model learning?** $\mathrm{R}^2$ of the intersection of $\psi$-classes is heavily affected when using counter-factual instances for the number of marked points $n$ and partitions $\boldsymbol{d}$. This indicates a strong causal connection between the partitions and the intersection numbers. This is also evident from the autocorrelation plot Figure 4, where the correlation between the intersection numbers is completely lost after disrupting the binding between instances and their partitions. The presence of such a strong causal relationship for $n$ and $\boldsymbol{d}$ is not surprising, but their mild impact on

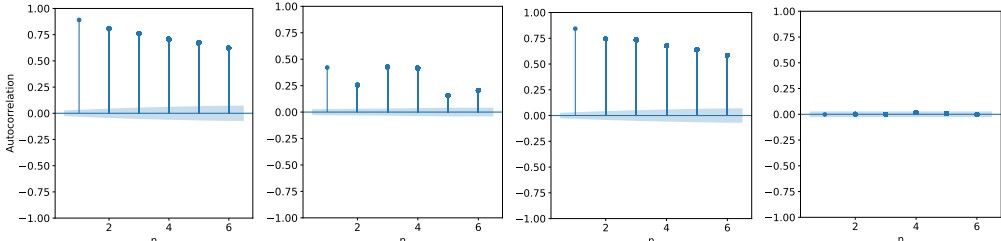

Figure 4: Autocorrelation plot across different number of marked points $n$ for clean (leftmost) and intervened runs (from left to right) for $n$, $\boldsymbol{B}$, and $\boldsymbol{d}$ respectively.

$R^2_{\text{probe}}$ is an interesting discovery. Despite a weak causal connection between $\boldsymbol{B}$ and the predicted intersection numbers, there is an evidence that *the exponential growth constant, $A = 2/3$, is causally connected to the quantum Airy structure initial datum $\boldsymbol{B}$*. This suggests that the network's hidden states of the $\boldsymbol{B}$ initial datum encodes this parameter. Meanwhile, the weak causal impact of $\boldsymbol{B}$ on the intersection numbers results in only a mild effect on their autocorrelation. This is expected since $\boldsymbol{B}$ contributes linearly, shown in Equation (2.4), while the dependence in $n$ and $d$ are factorial.

# 6 CONCLUSION

In this work, we tried to address a fundamental question: *Can Transformers perform and learn enumerative geometry and topological recursion?* To answer, we investigated a deep enumerative problem in algebraic geometry, specifically the computation of $\psi$-class intersection numbers. As a result, we introduced DynamicFormer, a multi-modal Transformer-based model, to predict the $\psi$-class intersection numbers. We analyzed the ability of the network to tackle such a task in a zero-shot setting. Capturing the recursive behavior of these invariants poses a non-trivial challenge for Transformers. Throughout our experiments, we identified the crucial role of non-linear activation functions and introduced a new one, Dynamic Range Activator (DRA), which improves prediction precision of recursive maps. **One important message of this work is that merely predicting or classifying a mathematical object with ML is insufficient. A prediction without proper uncertainty estimation is unreliable, especially in mathematics.** We enhance reliability by incorporating conformal uncertainty estimation and explainability for knowledge discovery.

In the second part of the results, we asked a simple question: *How exactly does the network perform enumerative geometry?* To answer this, we conducted a series of correlational, conformal, and causal interoperability analyzes. Firstly, by examining the internal vector space of the model, we discovered that the network is learning Virasoro constraints without them being imposed on the model. This finding is intriguing because it suggests that in cases where such relations and equations are unknown but are suspected to exist, a pre-trained Transformer could potentially aid in a human-machine collaboration by providing hints and evidence for these relations in a purely data-driven manner. We further explored whether it is possible to perform some form of abductive reasoning and hypothesis testing to estimate the parameters of asymptotic form for intersection numbers. Often, there is an expectation for such asymptotic growths that leads to a process of conjecture building and deductive reasoning to derive the final formula. In this work, we advocate that by using the network's internal understanding of the data, one can provide informed guesses and even reject alternative hypotheses about candidate parameters. To achieve this, we conducted a grid search on the exponential growth constant of the asymptotic limit for $\psi$-class intersection numbers, based on the network's hidden representation. Using probing techniques, we showed that this information is encoded non-linearly in the model's internal representation. We also performed the same analysis on the first few terms of the subleading polynomial terms. We found that these subleading terms are also encoded in a non-linear manner in the model's hidden space.

In the end, we analyzed the internal decision-making procedure of DynamicFormer using causal inference. By applying the causal tracing method with counterfactual interventions, we aimed to shed light on how various input modalities—namely, the partitions, quantum Airy structure data, and number of marked points—are causally responsible for the model's understanding of $\psi$-class intersection numbers and their large genus parameters.

ACKNOWLEDGMENTS

This research was supported by the Excellence Cluster ORIGINS, funded by the Deutsche Forschungsgemeinschaft (DFG, German Research Foundation) under Germany's Excellence Strategy – EXC-2094-390783311. A.G. acknowledges support from an ETH Fellowship (22-2 FEL-003) and a Hermann-Weyl Instructorship from the Forschungsinstitut für Mathematik at ETH Zürich. B.H. expresses his gratitude to Gaëtan Borot, Elba Garcia-Failde, Lukas Heinrich, and François Charton for their invaluable discussions throughout this work. We extend our thanks to the organizers of the TR Salento 2021 School and Workshop on Topological Recursion, where this project and the present collaboration were initiated. Finally, B.H wishes to thank Setareh Fadavi for her unconditional support.

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

## A  FUTURE DIRECTIONS

An interesting open question to explore with interpretability methods developed here, is the regime when $(g, n) \to \infty$ at the same time (so that $g/n$ is kept bounded above and below). At the moment, there is not even a conjecture as to what the asymptotic should behave like, and how it should depend on the partitions. Another fascinating avenue for future research is to explore recent developments in intersection numbers (Eynard & Mitsios, 2023). This new approach proposes a formula that involves sums over partitions of combinatorial factors, revealing how certain partitions with specific conditions have vanishing coefficients in their decomposition into elementary symmetric polynomials. Remarkably, empirical observations suggest that even more partitions have vanishing coefficients than the trivial ones. This unexpected result points to a deeper, hidden structure in intersection numbers, where the internal representation of the network could be very beneficial in examining these new results and uncovering the underlying patterns.

## B  RELATED WORKS

In recent years, a new frontier of ML applications, known as Artificial Intelligence for Mathematics (He et al., 2023), has emerged. This branch includes Automated Theorem Proving (Loos et al., 2017; Paliwal et al., 2020; An et al., 2024), Generation (Trinh et al., 2024; Wang & Deng, 2020; Lin et al., 2024), Autoformalization (Wang et al., 2020; Wu et al., 2022; Jiang et al., 2023; Mikuła et al., 2024; Ying et al., 2024), machine-human collaborative guidance (Bao et al., 2020; Anderson et al., 2021; Davies et al., 2021; Berman et al., 2022; Craven et al., 2023; Coates et al., 2023; Dong et al., 2024), examples/counterexamples search problems (Halverson et al., 2019; Gukov et al., 2021; Wagner, 2021; Gukov et al., 2023; Berglund et al., 2024; Romera-Paredes et al., 2024), and predictive problems (such as amplitude and mathematical expression prediction) (Ragni & Klein, 2011; Saxton et al., 2019; Dersy et al., 2023; Alnuqaydan et al., 2023; Belcak et al., 2022; d'Ascoli et al., 2022; Cai et al., 2024; Coates et al., 2024). These areas extensively use Transformers, particularly in problems arising from scattering processes.

## C  MATHEMATICAL BACKGROUND

In this section, we review some background on the original mathematical problem. Let $\overline{\mathcal{M}}_{g,n}$ be the moduli space of genus $g \geq 0$ stable algebraic curves with $n > 0$ distinct marked points. This is a complex, compact, smooth orbifold of dimension $3g - 3 + n$. Associated with each marked point is a natural cohomology class $\psi_i := c_1(\mathcal{L}_i)$, defined as the first Chern class of the cotangent line bundle $\mathcal{L}_i$ at the $i$-th marked point.

In the early 90s, Witten (Witten, 1991) analyzed a theory of two-dimensional topological quantum gravity, where $\psi$-classes play the role of observables, and the *intersection numbers* (sometimes called *amplitudes*)

$$\langle \tau_{d_1} \cdots \tau_{d_n} \rangle_{g,n} := \int_{\overline{\mathcal{M}}_{g,n}} \psi_1^{d_1} \cdots \psi_n^{d_n}, \qquad d_1 + \cdots + d_n = d_{g,n} := 3g - 3 + n, \qquad \text{(C.1)}$$

represent correlators of the theory. Based on a low genus profile and considerations from physics, Witten conjectured that the exponential generating function

$$Z(\boldsymbol{x}; \hbar) = \exp \left( \sum_{\substack{g \geq 0 \\ n > 0}} \frac{\hbar^{2g-2+n}}{n!} \sum_{\substack{d_1, \ldots, d_n \geq 0 \\ d_1 + \cdots + d_n = d_{g,n}}} \langle \tau_{d_1} \cdots \tau_{d_n} \rangle_{g,n} x^{d_1} \cdots x^{d_n} \right) \qquad \text{(C.2)}$$

satisfies an infinite tower of quadratic partial differential equations with respect to the formal variables $(x^i)_{i \geq 0}$ known as the Korteweg–de Vries (KdV) hierarchy. This was later proven by Kontsevich (Kontsevich, 1992).

In recent years, it has been discovered that many fundamental invariants in physics and geometry can be described via intersection numbers. In physics, interest in $\psi$-class intersection numbers has been revitalized due to their connection with Jackiw–Teitelboim gravity (Saad et al., 2019). From

an algebraic geometry perspective, $\psi$-class intersection numbers represent the most basic form of Gromov–Witten invariants (Behrend, 1997). These intersection numbers are central to enumerative problems addressed by (Kontsevich, 1992) and (Mirzakhani, 2007) regarding the symplectic volumes of the moduli space of metric ribbon graphs and hyperbolic Riemann surfaces. Computing intersection numbers is pivotal in the asymptotic counting of geodesic curves in both flat and hyperbolic random geometries (Mirzakhani, 2008; Delecroix et al., 2021; Andersen et al., 2023).

Given the wide and fascinating applications of intersection numbers, computing them for arbitrarily large genera remains an open problem. Their association with the KdV hierarchy has been illuminating for calculating these invariants (Witten, 1991; Kontsevich, 1992). Another approach to characterizing these numbers involves Virasoro constraints (Dijkgraaf et al., 1991), where the partition function is annihilated by a collection of differential operators that form a representation of the Virasoro algebra. Alternative recursive approaches include topological recursion (Eynard & Orantin, 2007), the cut-and-join equation (Alexandrov, 2011), and exact formulae for the generating series, such as error-function-type integrals (Okounkov, 2002) and the determinantal formula (Bergère & Eynard, 2009). However, the general computation of intersection numbers remains an open problem, as these methods can only be used to compute $\psi$-class intersection numbers for cases with small $g$ or $n$. Meanwhile, universality phenomena in flat and hyperbolic geometry and in the dynamics of surfaces manifest themselves in large genera. Only recently, Aggarwal (Aggarwal, 2021) has found a closed-form formula for the large genus asymptotic of $\psi$-class intersection numbers.

# D  DATA REPRESENTATION AND SETUP

The input data during training consisted of the sparse tensors $B$ and $C$ from Equation (2.5), the genus $g$, the number of marked points $n$, and the partitions $\boldsymbol{d} = (d_1, \ldots, d_n)$ of $d_{g,n}$. We did not include $A$ and $D$ as input, as they are fixed initial conditions for all $\psi$-class intersection numbers.

As we will consider values of $g$ and $n$ such that the dimension $d_{g,n}$ will never exceed $d_{\max} = 100$, the initial data will consist of only finitely many data. More precisely, the initial data during training were represented as follows.

- **$B$ Initial Data**: A rank 3 tensor $B_{i,j}^k \in \mathbb{R}^{d_{g,n} \times d_{g,n} \times d_{g,n}}$. Such a tensor links intersection numbers of the same genus (cf. Equation (2.4)). Since it is a sparse tensor with $d_{g,n} \leq d_{\max} = 100$, we chose to represent its non-zero components in the COO (Coordinate List) format. In the COO format, $B$ is represented as a set of 4-tuples, each containing the indices and the corresponding non-zero value:

$$\boldsymbol{B} = \left\{ (i, j, k, B_{i,j}^k) \mid B_{i,j}^k \neq 0 \right\} \tag{D.1}$$

  where $i, j, k \in [0, d_{g,n}]$, and $B_{i,j}^k$ are the non-zero components of the tensor. With this choice of $d_{\max}$, the maximum length of $\boldsymbol{B}$ is approximately 1500.

- **$C$ Initial Data**: A rank 3 tensor $C_i^{j,k} \in \mathbb{R}^{d_{g,n} \times d_{g,n} \times d_{g,n}}$. Such a tensor links intersection numbers of different genera (cf. Equation (2.4)). Again, since it is a sparse tensor with $d_{g,n} \leq d_{\max} = 100$, we chose to represent its non-zero components in the COO format. In the COO format, $C$ is represented as a set of 4-tuples, each containing the indices and the corresponding non-zero value:

$$\boldsymbol{C} = \left\{ (i, j, k, C_i^{j,k}) \mid C_i^{j,k} \neq 0 \right\} \tag{D.2}$$

  where $i, j, k \in [0, d_{g,n}]$, and $C_i^{j,k}$ are the non-zero components of the tensor. With this choice of $d_{\max}$, the maximum length of $\boldsymbol{C}$ is approximately 1500. In this setup, the $C$ initial datum are excluded. This is because the $C$-terms contributes quadratically, while the $B$-term contributes linearly. Hence, $B$ has a stronger effect on computing $\psi$-class intersections as confirmed in (Aggarwal, 2021).

- **Partitions**: The intersection numbers $\langle \boldsymbol{d} \rangle_{g,n}$ are labeled by partitions $\boldsymbol{d} = (d_1, \ldots, d_n) \in \mathbb{N}^n$ of $d_{g,n}$ of length $n$. An important feature of intersection numbers (and more generally, amplitudes computed from quantum Airy structures) is that they are invariant under permutation of the indices, that is, elements of the partition $\boldsymbol{d}$. For example given a partition $\boldsymbol{d} = (3, 0, 0)$, we have $\langle 3, 0, 0 \rangle_{1,3} = \langle 0, 3, 0 \rangle_{1,3} = \langle 0, 0, 3 \rangle_{1,3}$.

We trained the model using data up to genus $g = 13$ and then tasked it with predicting the intersection numbers for genera $g = 14, 15, 16$ and $17$.

# E    MODEL DETAILS

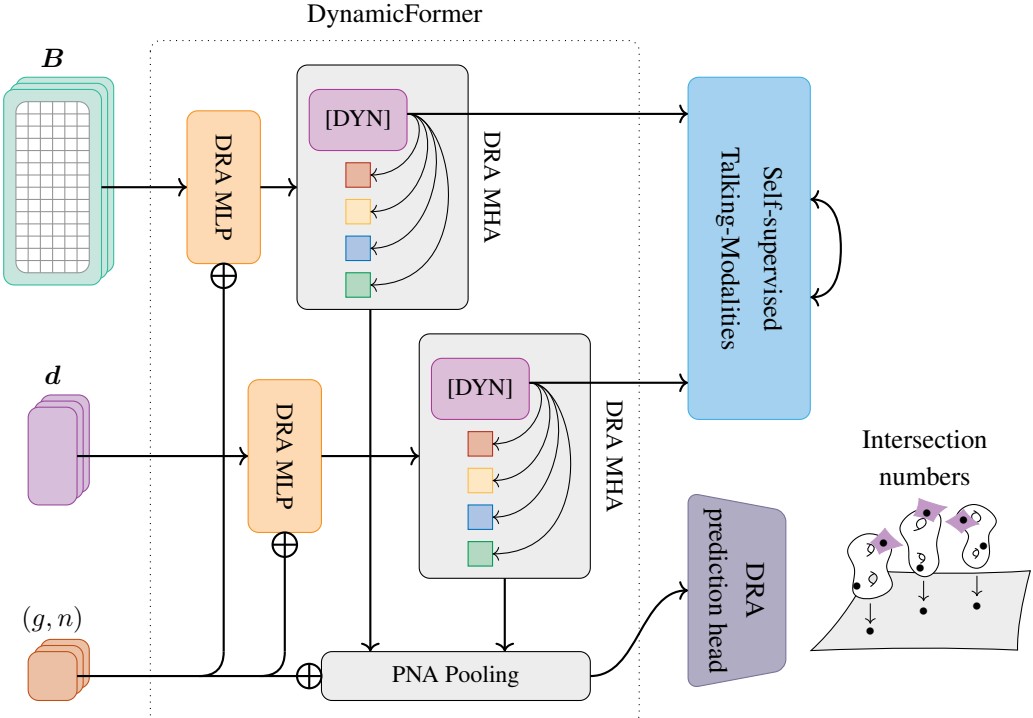

Figure 5: Illustration of DynamicFormer. It processes two main input modalities: the quantum Airy structure datum $B$ as an ordered sequence and partitions $d$ as a permutation-invariant set. The genus $g$ and number of marked points $n$ are incorporated as input properties, modulated with the main modalities at various stages. All layers, including the Multi-Head Attention (MHA) blocks, use the Dynamic Range Activator (DRA) non-linear activation function. The MHA block for $B$-modality integrates Dynamic Positional bias with linear positioning as well. The Talking-Modalities block consists of Batch Normalization (Ioffe & Szegedy, 2015) and the self-supervised loss. The DRA prediction head is a 2-layer MLP that predicts $\psi$-class intersection numbers in logarithmic scale.

To handle the different input modalities, i.e. the continuous tensor $B$ in COO sequence format and the discrete permutation invariant set $d \in \mathbb{N}^n$, we want to develop a multi-modal Transformer-based model. Transformers can effectively handle such structures due to their masked attention mechanisms and relative positional embeddings, which are adept at capturing the sparse structure of $B$. Moreover, the input $d$ is permutation invariant, and Transformers naturally accommodate this property through their self-attention mechanism that treats input elements symmetrically without imposing any ordering biases. Furthermore, Transformers are state-of-the-art models known for their flexibility and effectiveness in handling multi-modal data, which is crucial for our problem involving complex mathematical structures. In contrast, while Multi-Layer Perceptrons (MLPs) are universal approximators, they lack the inductive biases necessary to effectively handle the structured data and invariances inherent in our multi-modal input. They treat all input dimensions independently, making it challenging to leverage the sparse graph structure of $B$ or the permutation symmetry of $d$. This results in comparatively weaker scalability, especially when faced with the limited, complex, high-variance target distributions present in the $\psi$-class intersection numbers. Table 4 shows a comparison between MLPs and DynamicFormer.

The trunk of DynamicFormer consists of two modified Transformers: one discrete, acting on the embedding of the partitions $d$ and maintaining permutation equivariance, and one continuous, act-

| | Snake MLP | | DRA MLP | | DynamicFormer | |
|---|---|---|---|---|---|---|
| | $R^2 \uparrow$ | $CW \downarrow$ | $R^2 \uparrow$ | $CW \downarrow$ | $R^2 \uparrow$ | $CW \downarrow$ |
| | 69.3 | 8.44 | 76.5 | 7.73 | **95.8** | **3.42** |

Table 4: Comparison of $R^2$ and Conformal Width between models DynamicFormer and MLP with DRA and Snake non-linearity in the OOD regime.

ing on the $B$ tensor. The positions of elements in $B$, represented in COO format, correspond to particular combinations of indices that compute the intersection numbers. To enhance generalization for longer input sequences, we integrated a modified version of Dynamic Positional Bias (Wang et al., 2022; Zhang et al., 2024) with linear positioning. Dynamic Positional Bias computes a relative positional bias map for each attention head, which is learnable and adapts based on sequence length. Since it does not inherently account for the distance from the start or end of a COO sequence, we apply masking based on input sequence length.

After the Transformer trunk of the network, all the embedded information is aggregated by the Principal Neighbourhood Aggregation (PNA) layer (Corso et al., 2020), pooling the information from the modalities $(B, d)$. Once the information from these two branches, for each sample, is modulated with their corresponding genus and number of marked points positional attributes, $(g, n)$, it is mapped to the outputs, i.e., $\psi$-class intersection numbers, via an MLP head. DRA is the non-linear activation function used throughout these blocks. As the loss function, we use the Mean Absolute Error (MAE) loss. The total loss is, $\mathcal{L}_{\text{Total}} = \mathcal{L}_{\text{MAE}} + \mathcal{L}_{\text{TM}}$, where $\mathcal{L}_{\text{TM}}$ introduces a Self-Supervised Learning objective between the [DYN] registry tokens (Darcet et al., 2024) of each modality, inspired by Barlow Twins (Zbontar et al., 2021).

**[DYN] registry tokens and Talking Modalities.** In multi-modal learning, effectively sharing and integrating information across different modalities is crucial for building robust models. Inspired by the success of Self-Supervised Learning (SSL) (Gui et al., 2024), we introduce a simple modification to enhance correspondence between modalities. To achieve this, we concatenate dynamic register tokens (Darcet et al., 2024), [DYN], to each modality. These tokens are similar to [CLS] tokens in BERT (Devlin et al., 2019) and Vision Transformers (Dosovitskiy et al., 2021), in that they are passed through the attention layers alongside the input data. However, unlike [CLS] tokens, they are not directly used for the downstream supervised task of amplitude prediction. Instead, they are used indirectly, as register tokens (Darcet et al., 2024), which store and process global information, introduced for Vision Transformers. By attending to all tokens in the input, [DYN] tokens, similar to register tokens, do a soft contrastive learning update, enabling them to learn a global context for each sample. Unlike register tokens, [DYN] tokens are also optimized to ensure that the outputs from different modalities of data—specifically from the $B$ tensor, which provides global information per sample, and the partitions $d$, which serve as local information carriers—are as similar as possible while simultaneously reducing the redundancy of the information they carry. Their optimization is guided by the Canonical Correlation Analysis (Abdi et al., 2018) family of SSL, where the aim is to infer the relationship between two modalities by analyzing their cross-covariance matrices (Balestriero et al., 2023). Inspired by Barlow Twins (Zbontar et al., 2021), the two modalities start to have a "conversation", promoting an interaction in which the two modalities collaboratively refine their representations. The idea behind this is that such a consensus mechanism not only enhances the model's ability to generalize better to Out-Of-Distribution (OOD) data but also improves its robustness. The Talking Modalities loss is then defined as follows:

$$\mathcal{L}_{\text{TM}} = \sum_{i=1}^{k} \left(1 - \mathcal{C}_{ii}^{(B,d)}\right)^2 + \lambda \sum_{\substack{i,j=1,\ldots,k \\ i \neq j}} \left(\mathcal{C}_{ij}^{(B,d)}\right)^2, \tag{E.1}$$

where $\mathcal{C}$ is the cross-correlation matrix of the representations from the two modalities, $(B, d)$, $k$ is the dimensionality of the model's embeddings (vector representations), and $\lambda$ is the weighting factor that balances the contributions of talking modalities. The first term penalizes the diagonal elements of the cross-correlation matrix $\mathcal{C}$ for deviating from the identity, thus encouraging invariance. The second term penalizes the off-diagonal elements, encouraging the components of the representation

vectors to be decorrelated. Table 5 shows that this modification slightly enhances the performance of the model in the OOD prediction regime.

| | None | | [DYN] | | [DYN]+TM | |
|---|---|---|---|---|---|---|
| $R^2 \uparrow$ | $CW \downarrow$ | $R^2 \uparrow$ | $CW \downarrow$ | $R^2 \uparrow$ | $CW \downarrow$ |
| 91.1 | 4.91 | 94.6 | 4.05 | **95**.**8** | **3**.**42** |

Table 5: Comparison of $R^2$ and CW between models with and without [DYN] register token and Talking Modalities. A small performance improvement can be observed as a result of using register tokens Talking Modalities in the OOD regime.

## F    COMPARISON BETWEEN TRUE AND PREDICTED INTERSECTION NUMBERS

Figure 6 shows the numerical comparison between the true $\psi$-class intersection numbers and DynamicFormer's predictions, illustrating its performance in capturing the recursive behavior.

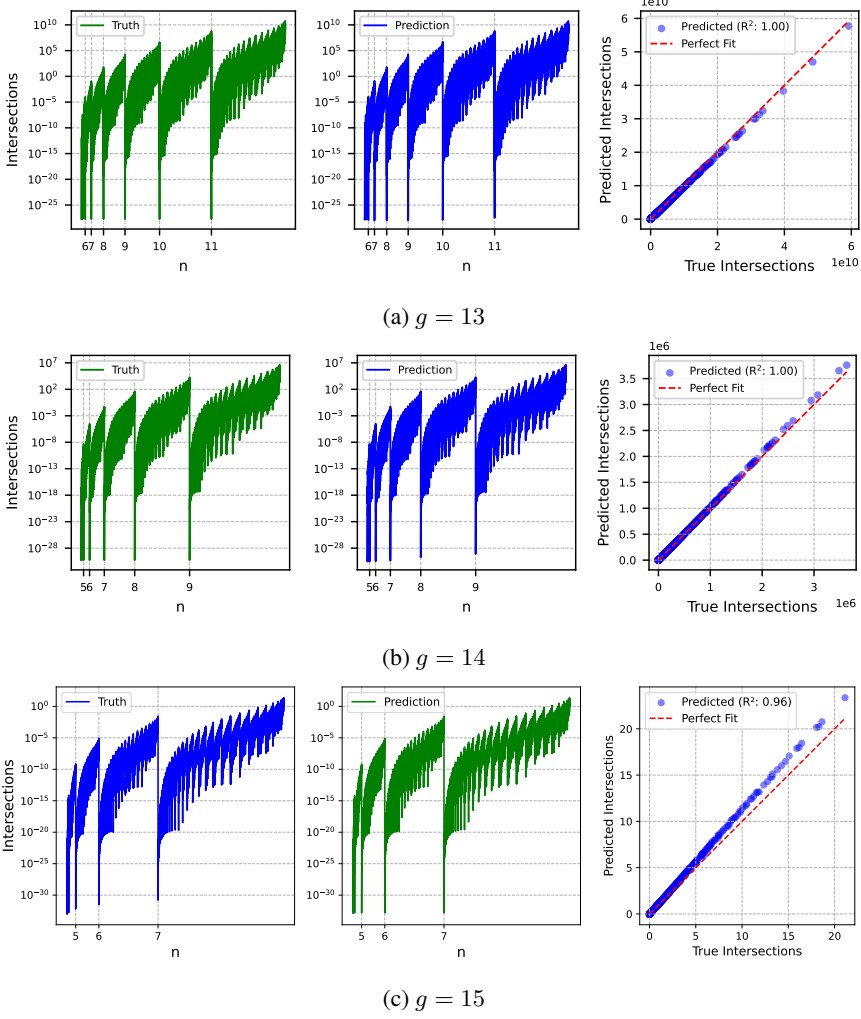

(a) $g = 13$

(b) $g = 14$

(c) $g = 15$

Figure 6: Comparison of true, predicted, and $R^2$ values for $g = 13, 14, 15$.

## G  UNCERTAINTY QUANTIFICATION: CONFORMAL PREDICTION

To estimate the uncertainty of the prediction, we incorporate Conformal Prediction (CP) (Vovk et al., 1999). CP is a probabilistic uncertainty quantification technique that provides prediction intervals in finite samples without making any distributional assumptions. We integrate the Inductive Conformal Prediction (Papadopoulos et al., 2007) with a dynamic sliding window (Xu & Xie, 2023) that, given the high heteroscedasticity in the data, computes rolling residuals for predicted intersections for each partition of equivalent marked points $n$. In our modification, we fit a quantile regression model to the residuals within a rolling window of equivalent marked points. This ensures that the residuals are more scale-dependent and appropriately adjusted for the inherent variability in the data, potentially capturing trends and heteroscedasticity better. As a result, we also report the empirical coverage and Conformal Width (CW) in the log scale.

## H  PRINCIPAL COMPONENT ANALYSIS OF MODEL'S INTERNAL REPRESENTATION

If we perform Principal Component Analysis (PCA) on the hidden embedding of each sample, $\boldsymbol{x}_{g,n}$, we observe in Figure 7 that DynamicFormer shows hierarchical representations of intersection numbers across both the genus $g$ and the number of marked points $n$. It seems that the model has indeed learned a recursive structure within the data.

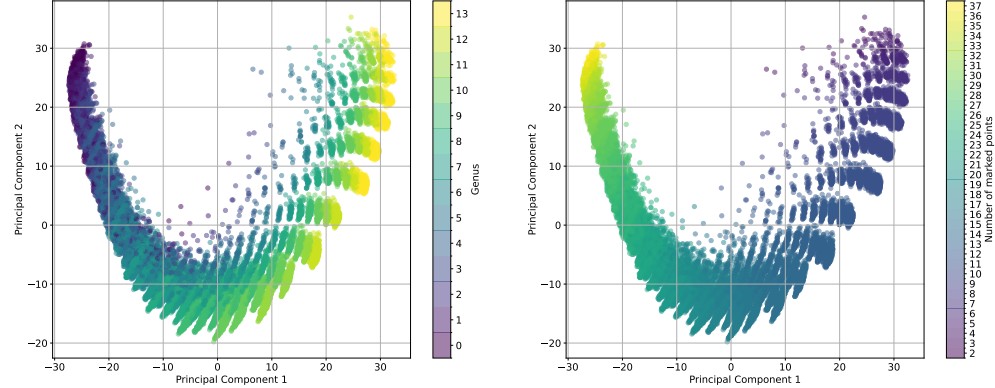

Figure 7: The first two principle components of model's hidden states across different genera (left) and number of marked points (right).

