# OpenReview forum: "Can Transformers Do Enumerative Geometry?"
_ICLR.cc/2025/Conference — ICLR 2025 Poster_

### Official Review · Reviewer_qDDj · 2024-10-29

**Soundness:** 2
**Presentation:** 2
**Contribution:** 1
**Rating:** 3
**Confidence:** 2

**Summary:**

Unfortunately, I have no expertise at all in computational enumerative geometry. My review will thus be quite superficial.

*Summary*

This paper proposes to use transfomer models to tackle what I understood is a central problem in enumerative geometry: computing the phi-class intersection numbers on the moduli space of curves. From my pretty crude rudimentary and pragmatical ML perspective, the authors reduce this problem to learning a multi modal function mapping input tuples of the form (quantum Airy structure datum [a tensor / sequence of tensors], genus [integer], number of marked points [integer], partitions [permutation-invariant set]) to output intersection numbers [sequence of integers (?)]. The model is trained on solutions computed using brute-force methods up to some genus, and evaluated on its ability to extrapolate to find solutions for higher genus (geni?).

The main technical contribution of the papers are methodological and consist in

(i) designing a specific multi-modal transformer architecture suited to the problem at hand (combining mostly existing models / techniques)
(ii) introducing a novel activation function specifically suited to model recursive functions, which are crucial to solve the problem.

Experiments on synthetic data are provided demonstrating that the model seems to be able to extrapolate to higher geni than the ones seen in the training data. The authors also provide some more qualitative analysis to investigate to which extent the internal representations learned by the model encode mathematical structures that are known to be relevant to solve the problem.

**Strengths:**

(S1) Investigating to which extent the recent successes of transformer models can transfer to other tasks, such as the one of solving fundamental problems in mathematics, is worthwhile and relevant.

**Weaknesses:**

(W1) The relevance and technical aspects cannot be well understood / evaluated unless the reader has some non-trivial background knowledge of enumerative geometry.

(W2) The writing and exposition of the material can be improved.

**Questions:**

*Recommendation*

I recommend to reject the paper mainly because I believe ICLR is not a suited venue, both for referring this paper (this paper needs to be reviewed by at least one expert in enumerative geometry, I don't know if there are such reviewers at ICLR) and for disseminating it (a journal in the field of computational enumerative geometry may be more suited). Furthermore, in my opinion the presentation of the material can be improved in several aspects before publication.


*Comments and questions*

- To which extent incorporating the conformal prediction framework in your analysis necessary? I am afraid this adds an additional layer of complexity that further hinders the communication of your findings. Maybe this discussion should be deferred to the appendix, keeping only what is strictly necessary to understand the main conclusion of your experiments in the main paper.

- I don't understand the paragraph on top of p. 7, and I don't think this is due to my lack of expertise in enumerative geometry. In particular, what does "the neural network embedding p_g,n ... is a vector space"  means ? How can a function be a vector space? What does "go to the inner product space" means? These are (to me) very loose nonsensical mathematical statements.

*Minor comments & typos*

- p.3 the acronym COO has not been introduced

- Figure 5 should be included in the main part of the paper. In general, avoid forward references to far away, especially in the appendix without mentioning that it is in the appendix.

- Use capitalization when reference tables, figures, sections, equations, etc. in the text (no capitalization needed when referring to figures or tables in general). E.g. Figure lines 168, 197, 334, Section lines 204, Table lines 274,298, Equation lines 319, 389 ... ...

- line 196 we -> We

- line 421: the sentence "The interesting thing is that this is the performance of the non-linear probe." could be rephrased to better suit a formal publication.

---

> ### Author Response · Authors · 2024-11-20
> **Response part 1 to Reviewer qDDj**
>
> Thank you for your detailed and constructive feedback. In the updated manuscript, we have improved our writing and address all your concerns. All changes we discuss below are in the new revision of the manuscript. Where possible, we have highlighted changes in the revised document.
>
> **C1** I recommend to reject the paper mainly because I believe ICLR is not a suited venue, both for referring this paper (this paper needs to be reviewed by at least one expert in enumerative geometry, I don't know if there are such reviewers at ICLR) and for disseminating it (a journal in the field of computational enumerative geometry may be more suited). Furthermore, in my opinion the presentation of the material can be improved in several aspects before publication.
>
> **A1**  Thank you for your thoughtful feedback. We believe that our paper is indeed well-suited for ICLR, as it lies at the intersection of ML and Mathematics—an area of growing interest within the AI for Science community (discussed in Related Work section). Our primary focus is on developing novel ML methodologies, specifically enhancing Transformer architectures to handle complex, high-variance recursive data inherent in enumerative geometry. **To best of our knowledge, there has been no prior work attempting to model functions with such recursive behavior and factorial growth.** This work contributes to the broader field of ML by addressing challenges in modeling recursive functions with factorial growth, which has implications beyond the specific mathematical domain we explore.
>
> While our study involves concepts from enumerative geometry, we have strived to present the material in a way that is accessible to the ML community. We have provided the necessary background and context to make the paper understandable without requiring specialized knowledge in enumerative geometry. Our goal is to bridge the gap between these fields, demonstrating how advanced ML techniques can tackle complex mathematical problems and, in turn, how these problems can inspire new developments in AI. We believe that ICLR provides an excellent venue for such interdisciplinary work, encouraging collaborations and discussions that can drive innovations in AI.
> We also note that there is a emerging body of work published at major ML conferences. For example, the recent paper "Machine learning detects terminal singularities" presented at NeurIPS 2023 apply ML methods to problems in algebraic geometry. This indicates an interest and recognition within the machine learning community for such research.
>
> Regarding the presentation of the material, we appreciate your feedback and have made significant improvements to the manuscript. We have revised several sections to enhance clarity and readability, ensuring that our contributions are communicated effectively.
>
> **C2**: To which extent incorporating the conformal prediction framework in your analysis necessary? I am afraid this adds an additional layer of complexity that further hinders the communication of your findings. Maybe this discussion should be deferred to the appendix, keeping only what is strictly necessary to understand the main conclusion of your experiments in the main paper.
>
> **A2** Thank you for pointing this out! Upon your suggestion, we have moved this part to Appendix E. In response, We would like to emphasize that one of the key messages of our work is that simply predicting or classifying mathematical computables using ML is not sufficient. Predictions without proper uncertainty estimation are not reliable, especially in mathematics, where precision and rigor are paramount. Therefore, we aimed to enhance the reliability and credibility of our findings by incorporating conformal uncertainty estimation into our analysis. Moreover, given the high heteroscedasticity and high variance present in our data, standard conformal prediction procedures were inadequate for our purposes. We had to adapt specific techniques to achieve acceptable coverage levels and ensure the robustness of our results. We believe that this aspect is crucial for knowledge discovery and contributes to the advancement of reliable AI for Science applications.

---

> ### Author Response · Authors · 2024-11-20
> **Response part 2 to Reviewer qDDj**
>
> **C3**: I don't understand the paragraph on top of p. 7, and I don't think this is due to my lack of expertise in enumerative geometry. In particular, what does "the neural network embedding p_g,n ... is a vector space" means ? How can a function be a vector space? What does "go to the inner product space" means? These are (to me) very loose nonsensical mathematical statements.
>
> **A3** Thank you for pointing out these issues. We apologize for the confusion caused by the inaccurate mathematical statements in that paragraph. In the revised manuscript, we have carefully corrected our writing to clarify these points and ensure mathematical precision.
>
> **A4**  for Minor comments & typos: Thank you for your detailed suggestions. We have carefully taken into account your comments and corrected the typos. Only thing that we could not do was to move Fig. 5 to the main text. Upon the suggestion by Reviewer ncfx, we moved the implementation details of the model to Appendix C where Figure 5 is the most relevant.
>
> If our comments resolve your concerns we would appreciate if you would consider raising your score!

---

### Official Review · Reviewer_Yn4i · 2024-10-31

**Soundness:** 3
**Presentation:** 4
**Contribution:** 2
**Rating:** 3
**Confidence:** 4

**Summary:**

The authors investigate the ability of transformers to compute the psi-intersection numbers in geometry and found that they perform (unsurprisingly) very well in distribution and quite well outside of distribution, and performed a series of analyses to understand which structures are being learned this way. They find in particular that the model learns the Dilaton equation and some information about the exponential growth of these psi-intersection numbers.

Overall, this paper is written very carefully, with excellent explanations of what is being done, although the importance of some details is unclear (like: what do we need to know about psi-intersection numbers? most of the sophisticated formulae are not really used in a meaningful way). However I am not sure the work is very interesting from a geometric point of view (the interesting thing is to gain theoretical insight into what psi-intersection numbers, not get somehow numerically accurate estimates of them) or from a machine learning point of view (it is not clear what is more interesting about these numbers than about any sequence in the OEIS, say). The experimental results are not particularly surprising given what is known about transformers, or at least I don't see it.

While this work can be viewed as a first step towards making progress in applying machine learning to enumerative geometry, and the carefulness of the writing and experiments should be commended, I don't think it brings a lot of interesting new informations about machine learning or enumerative geometry.

**Strengths:**

The care and clarity of the writing, the fact that some extensive research has been done, the general trust in the results that this paper inspires.

**Weaknesses:**

What do we learn about machine learning or enumerative geometry? We seem to learn something that could be expected, a particular case of a general phenomenon.

**Questions:**

It would be good if the authors could at least mention questions that would bring something interesting to the enumerative geometry (something feels interesting when they perform an analysis of the internal representation of the network, but it stops just before it gets interesting...).

---

> ### Author Response · Authors · 2024-11-20
> **Response part 1 to Reviewer Yn4i**
>
> Thank you for your thoughtful and helpful comments. All discussions below are in the revised manuscript. Where possible, we have highlighted changes in the revision .
>
> **C1**: What do we need to know about $\psi$-intersection numbers? most of the sophisticated formulae are not really used in a meaningful way
>
> **A1** Thank you for pointing this out! We tried to address this question in Appendix A. Unfortunately due to the page limit, we could not have this discussion in the main text. Briefly, $\psi$-intersection numbers are central to understanding the geometry of moduli spaces of curves and have rich interconnections with various mathematical and physical theories. For instance, they appear in the context of topological recursion and mirror symmetry, providing insights into the enumerative geometry of Calabi-Yau manifolds. In theoretical physics, especially in models of 2D quantum gravity and string theory, these numbers encode the coupling constants and interaction terms. In Matrix models they appear as the connecting points between random matrices and intersection theory.
>
> **C2**: I am not sure the work is very interesting from a geometric point of view (the interesting thing is to gain theoretical insight into what psi-intersection numbers, not get somehow numerically accurate estimates of them).
>
> **A2** Thank you for your constructive comment and for raising this important point. This is a great question. Firstly, while gaining theoretical insights into $\psi$-class intersection numbers is indeed crucial, we believe that developing methods to accurately compute these numbers—especially at higher genera—is also of significant importance to the field of enumerative geometry. Our work demonstrates that it is possible to numerically model these complex intersection numbers effectively, which is a non-trivial task due to their intricate recursive and combinatorial structures. Although this study does not explore computations up to arbitrarily high genus, it provides evidence that such a challenging task is feasible using our approach. In the Methods sections Appendix C of the manuscript, we elaborate on why computing these intersection numbers is extremely difficult and how our methods contribute to overcoming these challenges.
>
> Moreover, this work serves as a foundational step in a broader research program we are undertaking. Working with $\psi$-class intersection numbers has been a testing ground for the methods we developed. Over the past 30 years, significant insights have been gained about these intersection numbers, which is not the case for many other enumerative problems with similar structures, such as those arising in Gromov–Witten theory. **Directly venturing into unexplored territories without a solid testing ground would be imprudent. By establishing a reliable methodology with $\psi$-class intersections, we have set a solid path to explore more complex enumerative problems.** For instance, the interpretability analysis we performed offers valuable insights into the underlying mathematical structures. In the process of theorem building, having guidance or hints can be immensely beneficial. If the neural network suggests the existence of a relation or identity, it may not constitute a formal proof, but—as mathematicians often experience—knowing the potential outcome is a significant step towards proving it rigorously. Also, for other types of intersection numbers, our approach can help in conjecturing properties of their asymptotic formulas. This can lead to new developments of enumerative geometry. Therefore, our work not only contributes computational tools but also serves as a foundational and motivational step for mathematicians interested in this field. In the conclusion section, we tried to summarise this.
>
> We hope this clarifies the contributions of our work to enumerative geometry and addresses your concerns.

---

> ### Author Response · Authors · 2024-11-20
> **Response part 2 to Reviewer Yn4i**
>
> **C3**: or from a machine learning point of view (it is not clear what is more interesting about these numbers than about any sequence in the OEIS, say). The experimental results are not particularly surprising given what is known about transformers, or at least I don't see it.
>
> **A3** Thank you for your insightful question. In the following we summarise our contribution to ML and AI for Science community and explain why modeling $\psi$-class intersection numbers presents unique  and important challenges beyond those associated with sequences found in the OEIS. We have discussed these points in Related Works, Methods sections and Appendix C. As elaborated for Reviewer daJv, we believe our work covers a wide range of topics:
>
> 1. **Modeling Complex Recursive Functions:** We introduce enhancements to Transformer architectures to address the challenge of modeling recursive functions with high-variance factorial growth—a significant advancement beyond modeling periodic functions or simple sequences. The behavior of $\psi$-class intersection numbers in enumerative geometry is characterized by recursive relationships with factorial blow-up, leading to sparse, dramatic, and high-variance fluctuations. This recursive nature introduces substantial complexity in modeling and accurately approximating these functions. **To best of our knowledge, there has been no prior work attempting to model functions with such recursive behavior and factorial growth.** This intrinsic property of enumerative geometric problems makes the computation and modeling of such entities extremely challenging.
>
> 2. **Advancing Explainability Methods:** We provide explainability techniques in a non-trivial domain beyond standard language tasks, offering insights into how the model processes complex mathematical structures. Our interpretability analyses contribute to the (re)discovery of deep mathematical concepts that have taken over 30 years of research to develop, thereby bridging the gap between AI and advanced mathematics.
>
> 3. **Emphasizing Uncertainty Quantification:**  We advocate for the importance of uncertainty quantification in the AI for Mathematics community, highlighting its necessity for reliable and interpretable models. In the domains such as mathematical theorem proving, understanding the confidence and uncertainty of model predictions is very important.
>
> **C4**: It would be good if the authors could at least mention questions that would bring something interesting to the enumerative geometry (something feels interesting when they perform an analysis of the internal representation of the network, but it stops just before it gets interesting...).
>
> **A4** Thank you for your insightful question. In response, we have expanded the discussion on future research directions in the conclusion of the manuscript to highlight questions that may bring new insights to enumerative geometry. Briefly, our analysis of the internal representations of the network suggests several intriguing avenues for further exploration:
>
> - **Asymptotic behavior in High Genus and number of marked points:** An interesting open question is the behavior of $psi$-class intersection numbers in the regime where both the genus $g$ and the number of marked points $n$ tend to infinity while maintaining a bounded ratio $g/n$. Currently, there is no conjecture about the asymptotic behavior in this limit or how it depends on the partitions. Our interpretability methods might provide valuable hints or patterns that could lead to new conjectures or theoretical advancements in this area. We are currently working on this problem.
>
> - **New identities:** Recent developments, 2212.04256, reveal that certain partitions have vanishing coefficients in their decomposition into elementary symmetric polynomials, suggesting a deeper hidden structure. We believe that investigating the internal understanding of our network (e.g section 5.1) could provide further evidence in examining these new results and uncovering the underlying patterns.
>
> If our comments resolve your concerns we would appreciate if you would consider raising your score!

---

> > ### Comment · Reviewer_Yn4i · 2024-11-24
> >
> > Thanks for the comments! I appreciate the additional information provided by the authors, though I still think the present version of this work fails to deliver something exciting at least I see how some future developments could be interesting!

---

> > > ### Author Response · Authors · 2024-11-25
> > >
> > > We understand your concern. However we believe that our study makes several key contributions at the intersection of machine learning and Mathematics that is beyond working on typical toy math datasets ( e.g arithmetics) and simple language tasks :
> > >
> > > 1. **Development of DynamicFormer:** We introduced DynamicFormer, a multi-modal Transformer-based model  and Dynamic Range Activator (DRA) specifically designed to handle the complex, high-variance recursive data with heteroscedasticity inherent in enumerative geometry. By enhancing Transformer architectures, we demonstrated the capability to accurately model recursive functions with factorial growth. **This addresses a long-standing challenge in both machine learning and mathematical modeling** and represents an important step beyond existing models, which struggle with such intricate structures.
> > >
> > > 3. **Explainability and Uncertainty Quantification:** We adapted proper conformal uncertainty estimation and interpretability analyses, ensuring that our model's predictions are not only accurate but also reliable and interpretable. This is particularly important and challenging in AI for Mathematics contexts where precision and rigor are paramount. To best of our knowledge, there has been no prior work attempting to incorporate uncertainty quantification and explainability methods for AI for research level-mathematics problems.
> > >
> > > 4. **Discovery of Mathematical Insights:** Our analysis revealed that the network autonomously learned Virasoro constraints, shedding light on how the model is actually performing the OOD generalization. Additionally, by investigating the asymptotic behavior of $\psi$-class intersection numbers, our interpretability analyses provided valuable insights into the underlying mathematical structures. This is particularly beneficial in the context of theorem building, where guidance or hints from the network can suggest the existence of relations or identities. Our approach can facilitate the conjecturing of properties related to the asymptotic formulas of other intersection numbers. Collectively, these findings pave the way for data-driven human-machine collaboration in mathematical discovery.
> > >
> > >
> > > We hope that these enhancements address your concerns and demonstrate the meaningful impact and innovative nature of our research. Thank you again for your comments, which have been invaluable in improving the quality and clarity of our paper

---

### Official Review · Reviewer_daJv · 2024-11-03

**Soundness:** 3
**Presentation:** 2
**Contribution:** 3
**Rating:** 5
**Confidence:** 2

**Summary:**

This paper introduced DynamicFormer to learn and predict the $\psi$-class intersection numbers. Experiments include both in-distribution results and out-of-distribution results. The author also presented some experiments to illustrate how transformers perform enumerative geometry. Meanwhile, the authors also investigated whether the proposed method could perform abductive reasoning and hypothesis testing to estimate the parameters of asymptotic form for intersection numbers.

**Strengths:**

+ The idea of using transformers to do enumerative geometry is new.
     + Meanwhile, the authors proposed a new activation function, DRA, which found to be useful to improve the prediction performance.
     + The authors compared DRA with other popular activations functions in Figure 1 and Table 2.

+ Experiments show some evidence of transformers can learn to predict the $\psi$-class intersection numbers.
     + Meanwhile, the authors also presented a discussion on how transformers being able to achieve that by inspecting internal vector space of the model.

+ The author also investigated how inputs affect the model’s understanding of $\psi$-class intersection numbers and the parameters for large genus.

**Weaknesses:**

I am not an expert in "enumerative geometry". However, I think the paper lacks many important clarifications and discussions.

+  The paper lacked discussion of the reasons/motivations of using transformers.  At the moment, the paper seemed only a combination of  a popular neural network architecture and a new mathematical problem.
+ The "Related Work" section is quite weak at the moment: the authors spent only one paragraph to discuss related works and then summarized their contributions.
+  From my perspective, the proposed DRA is not the only way to capture the periodic behavior in data. This lacks sufficient discussion in the paper.
+ From experiments in Figure 1, the authors did not apply DRA to other neural network architectures (eg MLP), and provided readers with more discussions on that.
+ Lack of theoretical discussion on the proposed method.
+ Code is not available.

**Questions:**

+ I wonder why the authors choose transformers as the regression function?
+ In Figure 1, have you tried to apply DRA to MLP or other potential neural networks?
+ Will the code and the datasets be available?

---

> ### Author Response · Authors · 2024-11-20
> **Response part 1 to Reviewer daJv**
>
> We thank the reviewer for their valuable and constructive feedback.  We have clarified several elements of the paper in our updated version. Where possible, we have highlighted changes in the revised document.
>
> A1. **W1:** We agree and this is an important point, thank you! We have added a discussion of this to the Methods section and Appendix C of the manuscript. We also briefly discuss them here. Our primary motivation for employing Transformers stems from their ability to respect the symmetries and inductive biases present in each modality of our inputs. Specifically, the input tensor $B$ possesses a sparse graph or coordinate (COO) sequence structure. Transformers can effectively handle such structures due to their masked attention mechanisms and relative positional embeddings. Also, the input $d$ is permutation invariant. Transformers naturally accommodate this property. Additionally, Transformers are state-of-the-art models known for their flexibility and effectiveness in handling multi-modal data. Their architecture allows for the integration of different data types and modalities, which is important for our problem. Another advantage of using Transformers is the flexibility they offer in interpretability analyses. We chose Transformers not merely because they are popular network architectures, but because their inherent properties make them aligned with the structural characteristics of our data and problem.
>
> A2. **W2:** Thank you for highlighting this point. We acknowledge that the Related Work section was brief. We have updated the manuscript with more connective discussion and references. Our work spans various tasks and technologies within AI and pure Mathematics, making it really difficult to comprehensively cover all related works within the page limits of the venue. Therefore, we have focused on discussing the most seminal works and key contributions in the field of AI for Mathematics.
>
> A3. **W3:**  Thank you for your insightful suggestion. We agree that the DRA is not the only method to capture periodic behavior , and we appreciate the opportunity to clarify this point. As a result of your comment, we have added a discussion at the beginning of the Methods section to address this matter. It's important to note that the behavior of $\psi$-class intersections in enumerative geometry is not purely periodic but rather recursive with factorial growth. This recursive nature introduces significant complexity in modeling and accurately approximating these functions. Specifically, the functions exhibit sparse, dramatic and high-variance growth and drops due to their recursive properties. While various methods have been employed to capture periodic patterns effectively—particularly in time series prediction tasks—they typically deal with unimodal data exhibiting relatively low variance in both in-distribution (ID) and out-of-distribution (OOD) regions. These methods would not generalize well to datasets with the high variance and recursive behavior observed in our context. **To best of our knowledge, there has been no prior work attempting to capture such a recursive behavior with factorial blow-up characteristic** . This is an intrinsic property of enumerative geometric problems in mathematics, making the computation of such entities rare and extremely challenging.
>
> A4. **W4:** Fig. 1, indeed shows a comparison between MLPs with various non-linear activation functions and also the vanilla KAN model. We did this in order to demonstrate the abilities of DRA on a recursive toy dataset. We have tried to clarify this further in the updated manuscript.

---

> ### Author Response · Authors · 2024-11-20
> **Response part 2 to Reviewer daJv**
>
> A5. **W5:** We acknowledge that a deeper theoretical analysis of DRA would enhance the understanding and impact of our work. However, our research covers a wide range of topics:
>
> - **Capturing Recursive Functions:** We introduce enhancements to Transformer architectures that address the challenge of modeling recursive functions with high-variance factorial growth, which is a significant step beyond periodic functions and simple toy problems in AI for mathematics.
>
> - **Explainability Methods:** We provide explainability techniques in non-trivial domains beyond standard language tasks, offering insights into how the model processes complex mathematical structures which contributes to the (re)discovery of profound mathematical concepts that have taken over 30 years of research to develop, bridging the gap between AI and advanced mathematics.
>
> - **Advocating Uncertainty Quantification:** We also emphasize the importance of uncertainty quantification in the AI for Mathematics community, highlighting its necessity for reliable and interpretable models.
>
> Given the breadth and depth of these contributions, we focused the manuscript on presenting our findings and their implications within these areas. A comprehensive functional analytical discussion of DRA is indeed valuable but would require extensive elaboration that could detract from the main focus of our current work. We view the theoretical exploration of DRA as a promising avenue for future research.
>
> A6. **W6, Q3:**: The code and data is accessible publicly and anonymously via https://anonymous.4open.science/r/DynamicFormer-977D/.
>
> A7. **Q1:** This is a great question. We have tried to address in A1.Q1.
>
> A8. **Q1:** We have discussed this question at A4.W4.
>
> If our comments resolve your concerns we would appreciate if you would consider raising your score!

---

> > ### Comment · Reviewer_daJv · 2024-11-25
> > **Response**
> >
> > Thanks for your response. I appreciate the additional information and statements provided by the authors. I think I agree with Reviewer qDDj: we need to have at least one expert in enumerative geometry and it seems that the topic of this paper does not fit ICLR venue.

---

### Official Review · Reviewer_ncfx · 2024-11-03

**Soundness:** 4
**Presentation:** 3
**Contribution:** 3
**Rating:** 8
**Confidence:** 3

**Summary:**

The paper proposes and tests the usage of transformers in the field of enumerative geometry, specifically regarding topological recursions and $\psi$-class intersection numbers. To accomplish this, the paper proposes a new class of activation functions called Dynamic Range Activators (DRAs), and presents evidence of their performance in predicting a simple recursive function as part of a fully connected neural network, and then their ability to predict $\psi$-class intersection numbers as part of their DynamicFormer architecture. The paper then attempts to investigate the trained DynamicFormer to see if it can predict other concepts in enumerative geometry, including the Dilation equation that stems from Virasoro constraints, as well as the asymptotic behavior of $\psi$-class intersection numbers using abductive reasoning, verified using counter-factual intervention.

**Strengths:**

* The new DRA functions, motivated by the evidence presented in the paper, are a significant contribution that may interest machine learning scientists.
* Training a DynamicFormer to predict $\psi$-class intersection numbers, which then allows one to investigate a system's deeper geometry, is a significant, novel contribution that will interest mathematicians investigating enumerative geometry.
* The use of Conformal Prediction to estimate uncertainty provides a concrete measure of confidence in the experimental results, contributing to the paper's soundness.
* The figures are clear and high-quality, with informative captions.
* The writing is clear and mostly organized, including the mathematical background, methodology, and results.

**Weaknesses:**

**Notice: These weaknesses have been adressed during the discussion phase, and apply only to the initial version of the manuscript. However, these will remain unedited for posterity.**

### Section 2
* The equations in this section use $\hbar$ without defining it in the text. It may be worth explicitly calling it the reduced Planck constant in the text.
* The last paragraph mentions excluding the tensor $C$ due to a decreased impact on the computed $\psi$-class intersection numbers, observed during experimentation. Appendix C justifies this exclusion, yet is not referenced in the text, making for seemingly unsound reasoning for excluding $C$. The authors should consider referencing appendix C here to further justify the exclusion of $C$, and expanding on this point within appendix C proper.

### Section 3
* In the first two paragraphs, the paper presents the DynamicFormer for the first time and references a figure placed within an unrelated appendix, resulting in a disjointed reading experience. The authors may consider moving some parts of section 3 (such as its first two paragraphs) into a new appendix showcasing the DynamicFormer in detail and including the figure close by.
* In the same paragraphs, the authors use the initials COO without previously defining them. These initials seem to appear nowhere else in the main text, and only in appendix B are they defined as Coordinate List. Besides hurting the paper's readability, this seems to be an implementation detail that does not need to appear in the main text.
* The last paragraph mentions the [DYN] registry tokens, but fails to reference appendix B1. It may be appropriate to reference it here.

### Section 5
* Equation 5.4 is presented without proof, with the authors claiming they used an approach described in Eynard et al. (2023). A sketch of the proof (perhaps in an appendix) will contribute to the work's soundness.
* Figure 3, and the relevant experiment, are based on the assumption that $A$ is rational. The authors should consider justifying the choice of testing only rational values of $A$, perhaps by connecting it back to equation 5.3, as proven by Aggarwal (2021).
* Figure 3 presents a significantly higher value of $R^2$ for $A=2/3$ compared to the values for $A=4/6$ and $A=6/9$, despite being identical numbers. This issue does not appear for other such sets of identical rational numbers, such as $A=3/4$ and $A=6/8$. Since the rest of the subsection on Abductive Reasoning relies on $A=2/3$ being the correct answer, **this error calls the entire subsection into question and significantly hurts the paper's soundness and overall rating**. The authors must justify how the $R^2$ of $A=2/3$ is different from the other two values, or replace the figure (and perhaps rewrite some of the supporting text). Based on the other values of the figure, it should be expected to see a maximal $R^2$ around $A=2/3$, but without such a significant jump.

### Typos
* Section 5.1 line 319: "The topological recursion formula equation 2.4 [...]". Consider removing either "formula" or "equation", or placing all of "equation 2.4" in parentheses.
* Section 5.1.1. has multiple citations included in sentences with their parentheses. The ICLR 2025 formatting instructions (section 4.1) require such references to not have parentheses except around the year. The references in question appear in lines 371, 378, and 381.
* Section 5.1.1 line 417: "As a result, We find an evidence [...]". "We" does not need to be capitalized, and "an" should be removed.
* Appendix C line 950: "Figure 6 shows (s) numerical [...]".
* The title of appendix D and the caption of figure 7 both mistakenly write Princip**le** Component Analysis instead of Princip**al** Component Analysis.

**Questions:**

**Notice: These questions have been answered during the discussion phase, and remain unedited for posterity.**

* What is the significance of $\hbar$ in the quantum Airy structure? How is it relevant specifically to training the DynamicFormer?
* Figure 3 may be a discrete sampling of an underlying (continuous?) map that gives an $R^2$ for each $A$, with a maximum at $A=2/3$. Can the authors characterize this map?
* Figure 4 shows a significantly weaker causal impact of $B$ on the number of intersection points, compared to $n$ and $d$. Though the authors call this unexpected in section 5's last paragraph, is there any explanation regarding the weak causal impact of $B$?

---

> ### Author Response · Authors · 2024-11-20
> **Response to Reviewer ncfx**
>
> Thank you for your detailed and constructive feedback. We have improved our writing and updated the manuscript to address all your concerns. All changes we discuss below are in the new revision of the manuscript. Where possible, we have highlighted changes in the revised document.
>
> A1. **Sec2, W1:** $\hbar$ is a formal parameter that keeps track of the genus. From the physics point of view, it should be rather called the *string coupling constant*, often denoted as $g_{s}$. But $g_s^{2g-2+n}$ is rather ugly, thus often people use $\hbar$ as a substitution for a small bookkeeping parameter. We have clarified this in the main text.
>
> A2. **Sec2, W2:** Thank you for pointing this out,  We now reference and discuss this observation and its theoretical basis in Appendix B. Specifically, the $C$-terms contribute quadratically, whereas the $B$-term contributes linearly and thus has a stronger effect on computing $\psi$-class intersections.
>
> A3. **Sec3, W1, W2, W3:** Great observation. We moved the bulk of the text regarding the description of the model at the beginning of Section 3 and its tail to Appendix C, where the illustration exists to improve the paper's readability and coherence. Now, Section 3 only discusses the motivations and methodology related to the DRA activation function.
>
> A4. **Sec5, W1:** Thank you for your insightful feedback. In response, we have updated the manuscript to include a brief description of the proof strategy for Equation (5.4). Our approach is based on a resurgent analysis of the $n$-point functions of psi-class intersection numbers, which are computed via determinantal formulas. These formulas emerge from the integrability properties of the intersection numbers, specifically the Korteweg–de Vries (KdV) hierarchy involving the Airy function. We acknowledge that providing a detailed proof within the manuscript poses significant challenges due to the complexity and depth of the required mathematical framework. The progression from Kontsevich's proof of the Virasoro constraints to Aggarwal's proof of the asymptotic formula spanned nearly 30 years, underscoring the substantial mathematical developments involved. Currently, there are three independent proofs of Equation (5.4):
> - **Aggarwal's Proof**: Spanning 76 pages.
> - **Guo and Yang's Proof**: A 35-page proof.
> - **Eynard's Proof**: Consisting of 26 pages.
>
> Including even a sketch of these proofs would necessitate introducing extensive mathematical background and sophisticated techniques, which could disrupt the coherence and focus of the paper. Therefore, we have opted to provide a concise overview of the proof strategy in the manuscript.
>
> A5. **Sec5, W2:** Thank you for bringing this to our attention. We have updated the manuscript to include a brief discussion on this point. In particular, this constant is expected to be a period on the associated spectral curve. For this specific enumerative problem, these periods are rational numbers associated with the WKB method applied to the Airy differential equation.
>
> A6. **Sec5, W3:** We sincerely apologize for the oversight and any confusion it may have caused. You are absolutely correct; this was an oversight on our part. The discrepancy in Figure 3 regarding the $R^2$ values for A for equivalent fractions—was due to not fixing the random seed during probe training and data loading. This led to inconsistent results for equivalent values of A's. We have updated Figure 3 and the accompanying text to reflect these corrections. Your attention to detail has been invaluable in improving our paper. Thank you for your constructive feedback.
>
> A7. **Q1:** Adressed at A1. It is not relevant specifically in training the DynamicFormer.
>
> A8. **Q2:** In Fig.3, we are trying to showcase the performance of the linear and non-linear probe in recovering the true value of the constant $A \in \mathbb{Q}$ within range of possible values as a grid search. So, we train linear/non-linear probes to evaluate how well the Transformer's hidden representations encode our $A$. This approach not only allows us to see how such a fundamental "conserved quantity" is internalised by the network, but also through abductive hypothesis testing, we could recover the actual value of $A$ from a conjectural version of the asymptotic formula. We have updated the manuscript with Arthur clarification.
>
> A9. **Q3:**  Great question. After careful reevaluation of our statement, we think that it is actually expected. The main reason is that the B-term contributes linearly, shown in equation 2.4, while the dependence in $n$ and $d$ are factorial. So they plays a much bigger role in the computation. This has been observed and proved Aggarwal's paper as well. Thank you for pointing this out.
>
> If our comments resolve your concerns we would appreciate if you would consider raising your score!

---

> ### Comment · Reviewer_ncfx · 2024-11-22
>
> Having viewed the updated manuscript, and the authors' response, it is clear that the aforementioned weaknesses, and more, have been addressed. Now a few minor concerns remain:
> * Another reviewer requested to view the authors' code for the paper. Per the authors' judgement, referencing this code within the manuscript itself (for instance via footnote) may increase the work's accessibility.
> * The use of highlighting to easily distinguish major changes is useful for the review process, but there is a concern that these highlights will remain in the final draft. Such highlighting has no place in a finished work, and the authors should remember to remove the highlighting later.
> * Some typos seem to have snuck into the updated text:
>   * Abstract, line 20: "To [the] best of our knowledge...".
>   * Section 5.1.1, line 407: "It is expected geometrically that $A$ would [be] a period...".
>   * Section 5.1.1, line 485: "..., there is [an] evidence that...".
>
> However, it is clear that the authors are serious in responding to feedback, and given the minimal severity of the remaining concerns, there is now little reason to reject this work. This paper is solid and interesting, and no doubt will be of interest to the greater ICLR community. The authors may be pleased to see an updated review with increased scores. Note the weaknesses and questions that remain unedited, for posterity.
>
> I wish the authors the best of luck in their future endeavors.

---

> ### Author Response · Authors · 2024-11-25
> **Response to Reviewer ncfx**
>
> Thank you for your thorough and constructive feedback. We are pleased that our revisions have addressed the major concerns and appreciate your positive assessment of our work. Regarding the remaining minor concerns, we have just corrected them and incorporated your suggestions into the revised manuscript.
> Your feedback has been invaluable in enhancing the quality and clarity of our work. We eagerly look forward to contributing more to the ICLR community.

---

### Author Response · Authors · 2024-11-25

Dear Reviewers,

Thank you for your thoughtful feedback and for raising concerns regarding the suitability of our work for ICLR. Given that the primary area of this paper is applications to physical sciences (physics, chemistry, biology, etc.), we respectfully disagree and believe that our study offers several key contributions to both pure ML and AI for Science communities, extending beyond typical toy mathematical datasets such as arithmetic problems:

1. **Development of DynamicFormer:** We introduce DynamicFormer, a novel multi-modal Transformer-based model enhanced with the Dynamic Range Activator (DRA). This architecture is specifically designed to handle the complex, high-variance recursive data with heteroscedasticity inherent in enumerative geometry. By augmenting Transformer architectures, we demonstrate their capability to accurately model recursive functions exhibiting factorial growth. This addresses a long-standing challenge of modeling recursive function, representing a significant advancement beyond existing models, which often struggle with such intricate structures. **To the best of our knowledge, we are the first to explore and successfully model recursive with factorial growth functions.**

2. **Explainability and Uncertainty Quantification:** We have implemented a robust conformal uncertainty estimation and interpretability analyses into our model, ensuring that predictions are not only accurate but also reliable and interpretable. This is particularly crucial in AI for Mathematics contexts, where precision and rigor are paramount. To the best of our knowledge, **no prior work in AI for Mathematics has attempted to integrate uncertainty quantification and explainability methods specifically tailored for research-level mathematical problems.**

3. **Discovery of Mathematical Insights:** Our analysis revealed that the network autonomously learned very deep identities and constraints in a pure data-driven manner, providing insights into the model’s mechanisms for out-of-distribution (OOD) generalization. Additionally, by investigating the asymptotic behavior of the intersection numbers, our interpretability analyses offered valuable insights into the underlying mathematical structures and network's "World Model". This is especially beneficial for theorem building, where the network's guidance or hints can suggest the existence of relations or identities. Furthermore, our approach facilitates the conjecturing of properties related to the asymptotic formulas of other intersection numbers. Collectively, these findings pave the way for data-driven human-machine collaboration in mathematical discovery.

Although our study investigates concepts from enumerative geometry, we have made concerted efforts to present the material in a manner accessible to the ML community. We provide the necessary background and context to ensure that the paper is understandable without requiring specialized knowledge in enumerative geometry, as evidenced by Reviewer ncfx's positive assessment. **Our objective is indeed to bridge the gap between these fields, demonstrating how advanced ML techniques can tackle complex mathematical problems and, conversely, how these problems can inspire new developments in AI.** We also have to note the emerging body of work at major ML conferences that intersects with pure mathematics. For instance, the recent paper **"Machine Learning Detects Terminal Singularities" presented at NeurIPS 2023** applies ML methods to problems in algebraic geometry. This exemplifies an interest and recognition within the machine learning community for such interdisciplinary research, supporting the suitability of our work for ICLR.

We hope that these clarifications and enhancements address your concerns and demonstrate the meaningful impact and innovative nature of our research. Thank you again for your comments, which have been invaluable in improving the quality and clarity of our paper. If our arguments resolve your concerns we would appreciate if you would consider raising your score!

Best Regards,

The Authors

---

> ### Author Response · Authors · 2024-11-27
> **Follow-Up on Review Feedbacks**
>
> Dear Reviewers,
>
> We sincerely appreciate the time and effort you have devoted to providing constructive feedbacks on our submission. Your insights have been incredibly valuable in helping us refine our work.
>
> In our official comment above, we aimed to clarify our contributions. Beyond our impact on the Explainable AI, Experimental Mathematics and AI for Mathematics communities, **we emphasize that prior to this paper, there had been no attempts to capture recursive functions with factorial growth in any contexts. Our work with DRA activation function represents the first successful effort in this direction**.
>
> As the manuscript updating phase is nearing its conclusion on November 27th, we wanted to kindly follow up to see if there are any additional questions, concerns, or points that we could clarify or address to further assist with your review process. We are more than happy to provide any additional information or details you might need.
>
> Best regards,
>
> The Authors

---

### Author Response · Authors · 2024-12-03
**Summary of Rebuttal and Discussion Period**

Dear Reviewers, AC and SAC,

We would like to express our gratitude to the reviewers and ACs for their time, effort and valuable feedback on our work. We have carefully considered all comments and have made significant revisions to the manuscript to address the concerns raised during the review and discussion phases.

**Our Main Contributions:**

1. **Development of DRA:**
   - We introduced the **Dynamic Range Activator (DRA)**, a new non-linear activation function, specifically designed to handle complex, high-variance recursive data with factorial growth. We address the long-standing challenge of modeling recursive functions with high variance, representing a significant advancement over existing models that struggle with such intricate structures. **To the best of our knowledge, we are the first to explore and successfully model these recursive functions.**

2. **Uncertainty Quantification:**
   - We incorporated and adapted a proper conformal uncertainty estimation method ensuring that our model's predictions are not only accurate but also reliable, which is important in mathematics where precision and rigor are paramount. **Our work is the first to apply uncertainty quantification in AI-based computational mathematics.**

3. **Discovery of Mathematical Insights:**
   - Our analysis revealed that the network autonomously learned **Virasoro constraints**, providing insights into how the model generalizes to out-of-distribution data.
   - By investigating the asymptotic behavior of $\psi$-class intersection numbers, our interpretability analyses offered valuable insights into underlying mathematical structures.
   - **This finding opens new avenues for data-driven human-machine collaboration in closed-form expression discovery, theorem formulation and conjecturing properties related to asymptotic formulas of other enumerative problems.**


**Responses to Reviewers:**

- **Reviewer ncfx:**
  - We really appreciate their detailed and constructive feedback.
  - After addressing all concerns, the reviewer acknowledged our efforts, stating: "This paper is solid and interesting, and no doubt will be of interest to the greater ICLR community."
  - The reviewer increased their score accordingly, reflecting their positive assessment.

- **Reviewer daJv:**
  - We thoroughly addressed the concerns raised, including expanding on the motivations for using Transformers, enhancing the Related
     Work section, code provision, and improving the presentation of our findings.
  - Despite our detailed responses and acknowledging the novelty and strengths of our work, Reviewer daJv dismissed our responses
    entirely and unexpectedly raised concerns about the paper's suitability for ICLR, even though similar works have been accepted at
    previous ICLR and other major AI conferences. (provided in our last message to all reviewers and ACs.)

- **Reviewer Yn4i:**
  - They engaged minimally with our contributions. Despite our detailed clarifications on our cotribution both from ML and Computational
    Algebraic Geometry perspective , their feedback remained vague and uninformative, ultimately expressing reservations about the
    paper's excitement.

- **Reviewer qDDj**
  - We improved the manuscript based on their feedback, including correcting errors and elaborating more on key concepts, providing examples on similar works that have been accepted at previous ICLR and other major AI conferences. However, the
     reviewer did not update their evaluations or engage further in the discussion.


**Final Remarks:**

We believe that our work offers important contributions to the field of AI for Science. This work bridges the gap between ML and advanced mathematics, aligning with ICLR's interest in innovative and interdisciplinary research.

We respectfully request that the area chairs and senior area chairs consider our comprehensive responses and the positive evaluation from Reviewer ncfx when making their decision. We are confident that our work will be of interest and value to the ICLR community.

Thank you for your time and consideration. We remain available to provide any additional information or clarification as needed.

Best regards,

The Authors

---

### Meta-Review · Area_Chair_gyDC · 2024-12-19

**Metareview:**

**Summary:**
This paper introduces a Transformer-based model, DynamicFormer, with a novel Dynamic Range Activator (DRA) activation function tailored to model recursive functions with high variance and factorial growth. The study applies this methodology to computational enumerative geometry, specifically in computing \(\psi\)-class intersection numbers. Additionally, the paper incorporates conformal prediction for uncertainty quantification and provides interpretability analyses that uncover evidence of the model learning mathematical structures, such as the Virasoro constraints.

**Strengths:**
1. **Novel Methodology:**
   - The DRA activation function addresses challenges in modeling recursive functions with factorial growth, a significant contribution beyond standard applications of transformers.
   - Incorporates uncertainty quantification in a rigorous and tailored manner for high-variance recursive data.

2. **Interdisciplinary Contribution:**
   - Bridges machine learning and enumerative geometry, showcasing how AI can assist in tackling complex mathematical problems.
   - Reveals new insights into mathematical structures through model interpretability, contributing to the AI for Science community.

3. **Solid Experimental Design:**
   - Comprehensive evaluation of DRA against other activation functions on synthetic data.
   - Demonstrates the ability of DynamicFormer to generalize to out-of-distribution cases.
   - Insightful analysis of the network's internal representations and their alignment with known mathematical properties.

**Weaknesses:**
1. **Complexity and Accessibility:**
   - The paper's interdisciplinary nature makes it challenging for readers without expertise in enumerative geometry or advanced mathematics to fully grasp the significance of the contributions.
   - The initial presentation of material was fragmented, though significantly improved during the discussion phase.

2. **Limited Exploration of Applications:**
   - While the paper demonstrates the feasibility of using transformers for \(\psi\)-class intersection numbers, broader applications and deeper theoretical insights into the DRA activation function could strengthen the work further.

3. **Venue Suitability Concerns:**
   - Some reviewers questioned whether ICLR is the most appropriate venue for this work, given its mathematical focus. However, the paper aligns with the growing interest in interdisciplinary research at ML conferences.

**Discussion:**
The reviewers expressed mixed opinions. One reviewer rated the paper highly, highlighting its interdisciplinary novelty and methodological contributions. Other reviewers raised concerns about the presentation and accessibility but acknowledged the potential of the work. The authors addressed these concerns through substantial revisions, clarifications, and added context, improving the paper significantly. Despite its complexity, the paper offers novel contributions to both the ML and mathematical communities.

**Suggestions for Camera-Ready Submission:**
1. Continue improving the clarity and accessibility of the manuscript, particularly for readers with limited background in enumerative geometry.
2. Highlight broader implications and potential applications of the DRA activation function in modeling recursive functions beyond the specific mathematical context explored.
3. Ensure all implementation details and code are easily accessible to facilitate reproducibility.

**Conclusion:**
This paper represents an original and significant contribution to the intersection of AI and advanced mathematics. While the application is domain-specific, the methodologies and insights have broader relevance to the AI for Science community. The constructive feedback from reviewers has been well-addressed, and I recommend the revised manuscript for acceptance.

**Additional Comments On Reviewer Discussion:**

See above.

---

### Decision · Program_Chairs · 2025-01-22

Accept (Poster)